TOPICAL REVIEW

# On the mechanisms of brain blood flow regulation during hypoxia

Alexander Mascarenhas[1], Alice Braga[1], Sara Maria Majernikova[1], Shereen Nizari[1], Debora Marletta[2], Shefeeq M. Theparambil[1] [ID], Qadeer Aziz[1,3] [ID], Nephtali Marina[1,4] and Alexander V. Gourine[1] [ID]

[1]*Centre for Cardiovascular and Metabolic Neuroscience, Neuroscience, Physiology & Pharmacology, University College London, London, United Kingdom*

[2]*Main Library, University College London, London, United Kingdom*

[3]*Translational Medicine and Therapeutics, William Harvey Research Institute, Queen Mary University of London, London, United Kingdom*

[4]*Division of Medicine, University College London, London, United Kingdom*

The peer review history is available in the Supporting Information section of this article (https://doi.org/10.1113/JP285060#support-information-section).

*The Journal of Physiology*

**Abstract** The brain requires an uninterrupted supply of oxygen and nutrients to support the high metabolic needs of billions of nerve cells processing information. In low oxygen

conditions, increases in cerebral blood flow maintain brain oxygen delivery, but the cellular and molecular mechanisms responsible for dilation of cerebral blood vessels in response to hypoxia are not fully understood. This article presents a systematic review and analysis of data reported in studies of these mechanisms. Our primary outcome measure was the percent reduction of the cerebrovascular response to hypoxia in conditions of pharmacological or genetic blockade of specific signaling mechanisms studied in experimental animals or in humans. Selection criteria were met by 28 articles describing the results of animal studies and six articles describing the results of studies conducted in humans. Selected studies investigated the potential involvement of various neurotransmitters, neuromodulators, vasoactive molecules and ion channels. Of all the experimental conditions, blockade of adenosine-mediated signaling and inhibition of ATP-sensitive potassium ($K_{ATP}$) channels had the most significant effect in reducing the cerebrovascular response to hypoxia (by 49% and 37%, respectively). Various degree reductions of the hypoxic response were also reported in studies which investigated the roles of nitric oxide, arachidonic acid derivates, catecholamines and hydrogen sulphide, amongst others. However, definitive conclusions about the importance of these signaling pathways cannot be drawn from the results of this analysis. In conclusion, there is significant evidence that one of the key mechanisms of hypoxic cerebral vasodilation (accounting for ∼50% of the response) involves the actions of adenosine and modulation of vascular $K_{ATP}$ channels. However, recruitment of other vasodilatory signaling mechanisms is required for the full expression of the cerebrovascular response to hypoxia.

(Received 25 August 2023; accepted after revision 20 May 2024; first published online 6 June 2024)

**Corresponding authors** Nephtali Marina: Division of Medicine, University College London, London WC1E 6BT, UK. Email: n.marina@ucl.ac.uk

Alexander V. Gourine: Neuroscience, Physiology & Pharmacology, University College London, London WC1E 6BT, UK. Email: a.gourine@ucl.ac.uk

**Abstract figure legend** Schematic illustration of key hypothesized mechanisms mediating the cerebrovascular response to hypoxia, suggested by the results of studies included in this analysis.

## Introduction

The human brain consumes ∼20% of inspired oxygen at rest and up to 30% under conditions of increased mental and/or physical demand (Attwell & Laughlin, 2001; Howarth et al., 2012). The exceptionally high energy requirements of the brain make it susceptible to damage in conditions of insufficient supply of oxygen. If brain blood supply were to suddenly cease, the cerebral oxygen content (∼0.03 mM) would be sufficient to maintain neuronal activity for only a few seconds (Bailey, 2019; Buxton, 2010). Specific conditions that are potentially associated with reductions of brain oxygen supply, such as exposure to high altitude, lung disease or sleep apnoea, may have significant detrimental and lasting effects on brain function.

Efficient delivery of oxygen to brain cells is ensured by systemic mechanisms controlling the circulation and breathing (Spyer & Gourine, 2009), as well as by specific mechanisms that control brain tissue perfusion through an extensive network of cerebral blood vessels (Hoiland et al., 2016). Specialized oxygen sensors in the carotid and aortic bodies are sensitive to changes in the arterial oxygen, and in conditions of systemic hypoxia, they trigger adaptive increases in breathing, sympathetic activity and systemic arterial blood pressure (Prabhakar

**Alexander Mascarenhas** is a PhD student at University College London, supervised by Professor Alexander Gourine. He received an integrated master's degree from the University of Warwick, where he studied the properties of connexin hemichannels under the supervision of Professor Nicholas Dale. His current research focuses on the cellular and molecular mechanisms of oxygen sensing in the brain, as well as the physiological mechanisms underlying the control of cerebral blood flow during hypoxia.

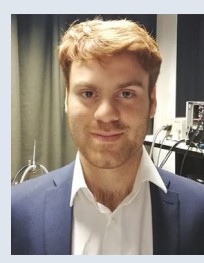

& Semenza, 2015). However, arterial oxygen sensors are located outside of the brain and therefore not able to detect brain tissue hypoxia or respond to any significant regional differences in brain oxygen supply (discussed in Gourine & Funk, 2017).

The mechanisms of *hypoxic cerebral vasodilation* maintain oxygen delivery to the brain by increasing cerebral blood flow (CBF) in response to hypoxia (Hoiland et al., 2016; Willie et al., 2014). These mechanisms may also be responsible for redistribution of cerebrovascular blood flow in accordance with the differences in regional brain tissue oxygen concentration. This is important because significant gradients of brain tissue partial pressure of oxygen ($P_{O_2}$) occur even at normal arterial $P_{O_2}$ and saturation (Devor et al., 2011; Kasischke et al., 2011), indicating that certain regions of the brain may experience significant reductions of oxygen supply (Beinlich et al., 2024; Gjedde, 2002; Ndubuizu & LaManna, 2007), and would need to recruit the mechanism(s) of brain tissue oxygen sensing capable of increasing CBF locally.

The physiological mechanisms responsible for hypoxic cerebral vasodilation have been the subject of significant research interest for the past 30 years. However, despite continued experimental scrutiny, the cellular and molecular mechanisms underlying the effects of hypoxia on brain vasculature are not fully understood. Here we report the results of a systematic review and analysis of published data obtained in experimental animal and human studies that investigated these mechanisms. Firstly, by analysing the hypoxic cerebrovascular responses recorded under control conditions, we aimed to evaluate the key differences in experimental design that could potentially explain the significant heterogeneity in the reported data. Secondly, by comparing the effects of pharmacological or genetic blockade of various signaling pathways on the expression of the hypoxic cerebrovascular response we determined the relative significance of the hypothesized mechanisms, suggested by the results of published experimental animal and human studies.

## Methods

This systematic review and analysis of the literature were conducted in accordance with the guidelines laid out in the Preferred Reporting Items for Systematic Reviews and Meta-Analyses (PRISMA) (Liberati et al., 2009), and the Systematic Review Protocol for Animal Intervention Studies (SYRCLE) (de Vries et al., 2015). This systematic review was not pre-registered.

### Study selection

*Types of studies.* Interventional pharmacological studies conducted in experimental animals and human participants and interventional studies on groups of animals with genetic manipulations designed to delete or alter specific genes and their respective control animals were included in the analysis.

*Types of participants.* The subjects of the study were mammals, including humans, without age or sex restrictions. The subjects and participants of the studies were healthy. Studies that included subjects with specific diseases (or animal models of disease) were excluded. Genetically modified animals were not considered to be disease models.

*Types of interventions.* In all the studies, a reduction in the fraction of inspired oxygen was used to induce arterial hypoxaemia and systemic hypoxia. This analysis included studies that examined changes in cerebrovascular flow in response to hypoxia under conditions of pharmacological (including administration of specific receptor antagonists, blockers of ion channels, inhibitors of enzymes responsible for biosynthesis of signaling molecules, or enzymes which facilitate degradation of signaling molecules in the extracellular space) or genetic (gene knockout) blockade of specific signaling pathways. The control groups included the same or separate subjects receiving an administration of a vehicle control or wild-type animals. Pharmacological studies that did not include treatments involving the administration of a vehicle solution were excluded. We only considered studies in which a vehicle or a pharmacological agent were administered prior to inducing hypoxia and in which hypoxia was the first experimental condition tested following the application of a vehicle or an agent or in conditions of genetic blockade of a specific signaling pathway.

*Types of outcome measures.* Our primary outcome measure was the percent reduction of the cerebrovascular response to hypoxia in conditions of pharmacological or genetic blockade of specific signaling pathways studied in experimental animals or in humans. We also analysed and compared the cerebrovascular responses to decreases in the level of inspired oxygen recorded in experimental subjects in control conditions (after administration of a vehicle control solution or in wild-type animals) and reported in the selected studies.

**Database search strategy.** Database searches using Medline, EMBASE + EMBASE Classic via Ovid and CINAHL Plus via Ebscohost were performed on 17 January 2024 by A.M. under the guidance of two investigators (D.M. and N.M). The list of terms used for the database searches are given in Tables S1–S3. The searches were conducted without any date restrictions but were limited to English language sources. All retrieved sources were then exported to Endnote and duplicate items were removed.

**Selection of studies.** Studies were selected for the analysis if they involved pharmacological and/or genetic blockade of specific signaling pathways and determined the effect of these treatments on hypoxia-induced changes in cerebrovascular flow recorded in human subjects or in *in vivo* experimental animal models. The title, the abstract and the main text of each identified study underwent independent screening by two investigators (A.M. and N.M.) using the Rayyan software (https://www.rayyan.ai/). Only full-length articles reporting original research data were considered. The following steps were then followed to select the studies for further analysis. (1) All duplicates were removed. (2) Studies that did not meet the inclusion criteria were excluded. In cases where the relevance of a particular study was not immediately obvious, the study was deemed eligible for full-text evaluation. (3) Full-text versions of all the selected sources were obtained and assessed against the same eligibility criteria. Any differences in opinion regarding article inclusion were resolved by reaching consensus between the authors. The number of excluded studies and the reasons for each study exclusion were recorded. (4) The articles that passed Step 3 were included in the pool of sources for data extraction.

**Data extraction.** The selected studies were evaluated and the following key information was extracted: publication year; subject characteristics, including species, age, weight and sex; sample sizes; the information on whether the experimental subjects were unanaesthetized or anaesthetized (if the latter, the details of the anaesthetic protocol); study methods, including the protocol of inducing hypoxia and the methods used to measure the cerebrovascular responses to hypoxia; intervention details, such as the pharmacological agent(s) used, the dose and the route of administration, or details of the genetic manipulation applied, including the gene(s) deleted or modified. In cases where the data obtained in experimental animal studies were reported at several time points and at graded stages of hypoxia, the largest difference in the hypoxia-induced cerebrovascular response between the control and experimental groups was taken as the effect of the treatment. For the analysis of data obtained in human studies, the values of cerebrovascular flow taken at 80% $S_{aO_2}/S_{pO_2}$ were extracted from the selected publications. This allowed comparisons of the response magnitude and the effects of treatments between different human studies. WebPlotDigitizer software (https://automeris.io/WebPlotDigitizer/) was used to extract and estimate the results of interventions from the charts, if the numerical data were not reported in the text.

**Risk of bias assessment.** For studies that included human participants, the Cochrane Collaboration tool for risk of bias assessment (Higgins et al., 2011) was used. For studies involving experimental animals, a custom-designed version of the SYRCLE tool (Hooijmans et al., 2014) was applied. These tools analyse the degree of bias due to: selection, performance, detection, attrition and reporting. The overall assessment of each study was classified as (1) 'low risk of bias' if none of the domains were rated as high and/or if fewer than three domains were rated as unclear, (2) 'unclear risk of bias' if more than three domains were rated as unclear and (3) 'high risk of bias' if at least one domain was rated as high risk. A risk-of-bias summary table was created in Review Manager version 5.4 (Cochrane Collaboration).

**Data analysis.** Mean values reporting the magnitude of the cerebrovascular response to hypoxia in the control and experimental conditions and the number of experimental subjects were retrieved from each source. By comparing the expression of the hypoxic cerebrovascular response recorded under control conditions, we evaluated the key differences in experimental design that could potentially explain the significant heterogeneity in the results reported in the selected studies. The data were grouped into three categories to evaluate the effects of the species of experimental subjects, anaesthesia and the methods used to measure the cerebrovascular responses.

Our main outcome measure was the percent reduction of the cerebrovascular response to hypoxia in conditions when the hypothesized signaling mechanism(s) were blocked either pharmacologically or genetically in studies involving experimental animal models or in human participants. From 34 studies included in the analysis, 20 reported data as changes in absolute measures of cerebrovascular reactivity (e.g. flow in mL/100 g/min, vessel diameter in $\mu$m, etc.), and 14 studies only reported the normalized data. Nineteen studies described the effect of the experimental treatment on the baseline variable; 15 studies did not assess or report changes in baseline cerebrovascular flow following pharmacological treatment or genetic manipulation. All human studies included in the analysis provide absolute values. For the analysis of results obtained in human studies, the data were extracted and the effect of an experimental treatment on the expression of the cerebrovascular response to hypoxia was calculated based on the absolute values and expressed as the percentage of the control response.

In the majority of selected animal studies, the absolute values were not reported and there was no mention in the text of the baseline changes. For the analysis and comparison of data obtained in animal studies, the effects of experimental treatments on the expression of the hypoxic response were calculated based on the reported normalized data. Means were taken from the data obtained in the control and experimental groups, and the

effect of a treatment was expressed as a percentage of the control response.

Forest plots were constructed to display the results obtained in the selected studies alongside the reported effects of each study treatment on the expression of the hypoxic vasodilatory response. If the effect of the experimental treatment on baseline flow was described or the data reported could be used to assess the differences in the baseline flow, the data were included in the forest plots.

The data were grouped into categories according to the main experimental condition targeting the signaling pathways mediated by adenosine, ATP-sensitive potassium channels, nitric oxide (NO), cyclooxygenase products, amongst others, and further categorized based on the specific pharmacological agent used or genetic manipulation performed. The individual data points were plotted in accordance with the targeted signaling pathway; the effect means were calculated and plotted with their respective 95% confidence intervals (CIs). Experimental animal studies and studies involving human participants were assessed separately. Data were meta-analysed if at least four separate studies targeting the same signaling pathway met the selection criteria for inclusion (Fu et al., 2011). The data were analysed using one-way ANOVA. Statistical analysis of the data was performed using OriginPro software (version 8).

## Results

**Identification of studies for inclusion.** Database searches using the specific terms listed in Tables S1–S3 returned 6062 articles. After removing 1214 records found to be duplicates, the titles and the abstracts of 4848 articles were screened. In total, 246 articles met the eligibility criteria for full-text screening. Following full-text review of these 246 sources, 28 research articles describing the results of experimental animal studies and 6 articles describing the results of investigations conducted in human participants were selected for data extraction and analysis (Fig. 1). The remaining 212 articles did not meet the selection criteria for inclusion because of one or more of the following reasons: study protocols did not involve pharmacological or genetic blockade of physiological mechanisms followed by hypoxia testing *in vivo* ($n = 126$), study protocols did not include control treatments involving the administration of a vehicle ($n = 35$), study protocols did not evaluate the effect of hypoxia on cerebrovascular flow *in vivo* ($n = 32$), publication type was not an original research article ($n = 29$), non-mammalian species were used ($n = 3$), full text was unavailable ($n = 3$), sample sizes were not reported ($n = 2$) or experimental conditions were as such that hypoxia induced cerebrovascular constrictions ($n = 1$) (Fig. 1).

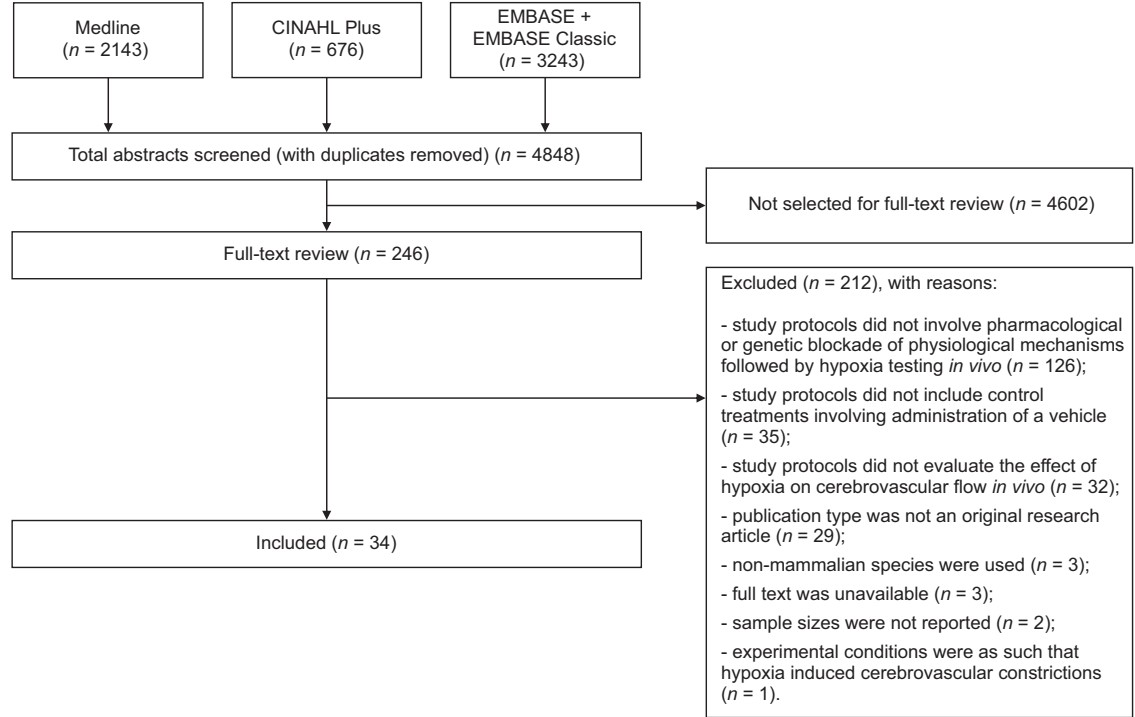

**Figure 1. PRISMA flow diagram illustrating the results of the literature search**
Thirty-four studies were selected for systematic review. PRISMA, Preferred Reporting Items for Systematic Reviews and Meta-Analyses.

## Characteristics of studies

*Human studies.* Six publications included in this systematic review reported the results of studies conducted in humans involving a total of 55 participants, with group sample sizes ranging between 5 and 12 (Bowton et al., 1988; Fan et al., 2011; Hoiland et al., 2017, 2023; Kellawan et al., 2020; Rocha et al., 2020). All studies recruited healthy men and women (average ages of participants in these studies were between 24 and 35 years old) with measurements of cerebrovascular flow performed in non-anaesthetised subjects breathing spontaneously. All studies reported increases in CBF in response to hypoxia in control conditions, i.e. following the administration of a vehicle solution. The test drugs were administered acutely and their effects on the hypoxic cerebrovascular response were evaluated. The studies were conducted in North America (5 publications) and South America (1 publication). The general characteristics of human studies included in this analysis are provided in Table S4.

*Experimental animal studies.* Twenty-eight publications included in this systematic review reported the experimental data obtained in a total of 563 animals, with group sample sizes ranging between 5 and 120. The most commonly used experimental subjects in the selected studies were piglets (13 publications), followed by rats (7 publications/8 studies), dogs (3 publications), rabbits (2 publications), mice (2 publications) and sheep (1 publication). Most studies (23 publications) were conducted in animals kept under general anaesthesia and mechanically ventilated (Armstead, 1995, 1998, 1999; Ben-Haim & Armstead, 2000; Christie et al., 2023; Kanu & Leffler, 2007; Kutzsche et al., 2002; Laudignon et al., 1990; Liu et al., 2015; McPhee & Maxwell, 1987; Miekisiak et al., 2008; Morii et al., 1987; Morikawa et al., 2012; Pelligrino et al., 1993, 1995; Shankar & Armstead, 1995; Simpson & Phillis, 1991; Taguchi et al., 1994; Tomiyama et al., 1999; Wagerle et al., 1983; Weiss & Buchweitz-Milton, 1988; Wilderman & Armstead, 1997, 1998). Spontaneously breathing conscious dogs and piglets were used in studies described in 5 publications (Audibert et al., 1991, 1995, 1998; Coyle et al., 1993, 1995). The most commonly applied anaesthetic protocol involved the use of ketamine/$\alpha$-chloralose/acepromazine (7 publications), followed by pentobarbital (5 publications), halothane (3 publications), fentanyl (2 publications/3 studies), isoflurane (2 publications), ketamine/$\alpha$-chloralose (1 publication), pentobarbital/fentanyl (1 publication), isoflurane/urethane/$\alpha$-chloralose (1 publication) and urethane/$\alpha$-chloralose (1 publication). In 16 studies, cerebrovascular responses to hypoxia were evaluated in separate groups of control (administration of a vehicle) and experimental (administration of a pharmacological agent) animals. In 12 studies, the hypoxia-induced cerebrovascular responses were first recorded following the administration of a vehicle (control response) and then after the administration of a pharmacological agent to block a specific signaling pathway in the same animal after a recovery period. Two studies compared the expression of the cerebrovascular response to hypoxia between the groups of transgenic (knockout) mice and the wild-type control animals. The studies were conducted in North America (24 publications), Europe (2 publications), Asia and North America (1 publication), and Oceania (1 publication). The general characteristics of experimental animal studies included in this analysis are provided in Table S5.

*Methods of cerebrovascular flow measurement.* Cerebrovascular responses to hypoxia were assessed by various experimental techniques including measurements of pial artery diameter (11 publications), radiolabelled microspheres (11 publications), duplex ultrasound (3 publications describing human studies), transcranial Doppler ultrasound (2 publications describing human studies), laser Doppler flowmetry (3 publications), two-photon imaging of cortical arteriole diameter (2 publications), xenon washout (1 publication describing a human study) and the indicator fractionation method (1 publication).

**Hypoxia-induced changes in cerebral blood flow.** In all the selected studies, the hypoxic cerebral vasodilatory response was induced by lowering the fraction of oxygen in the inspired gas mixture. Publications reported the level of hypoxia expressed as the fraction of inspired oxygen given and/or as values of arterial $P_{O_2}$ and saturation achieved when the concentration of inspired oxygen was reduced.

We first compared the hypoxia-induced cerebrovascular responses recorded under control conditions across all the selected studies. Our analysis revealed significant variations in the magnitude of the response ranging between 17% and 196% above the baseline. The data were then grouped into three categories to evaluate the effects of the species of experimental subjects (Fig. 2*A*), the type of anaesthesia (Fig. 2*B*) and the method used to measure the cerebrovascular response to hypoxia (Fig. 2*C*).

The largest cerebrovascular responses to hypoxia were recorded in studies involving anaesthetized sheep (increase by 102%), whilst the smallest average responses were recorded in human studies, involving measurements of CBF using transcranial Doppler ultrasound, duplex ultrasound or xenon washout methods (increases in CBF by $27 \pm 3\%$; mean $\pm$ SEM) (Bowton et al., 1988; Fan et al., 2011; Hoiland et al., 2017, 2023; Kellawan et al., 2020; Rocha et al., 2020). Hypoxia-induced cerebrovascular responses varied depending on the type of anaesthetic

agent used, ranging from 27% in piglets anaesthetized with pentobarbital/fentanyl (Kutzsche et al., 2002) to $130 \pm 23\%$ on average in animals (rats, piglets, sheep and rabbits) anaesthetized with pentobarbital.

**Signaling mechanisms mediating cerebrovascular response to hypoxia.** We next evaluated the significance of signaling mechanisms that were hypothesized to mediate the cerebrovascular response to hypoxia based on the results reported in the selected publications. The data are expressed as the percent reduction of the cerebrovascular response to hypoxia recorded in experimental animal and human studies that targeted 12 signaling mechanisms either pharmacologically or genetically.

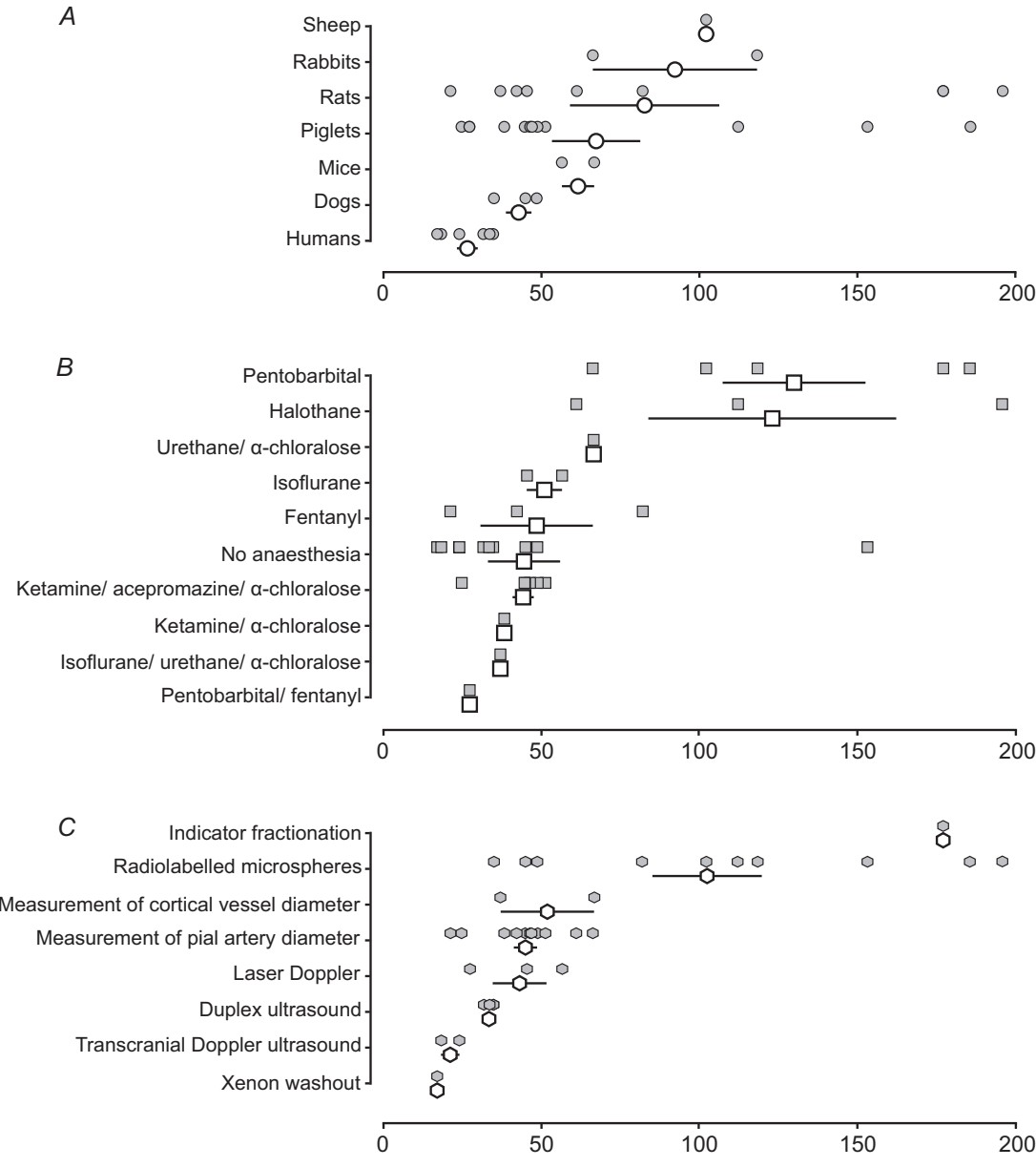

**Figure 2. Expression of the hypoxia-induced cerebrovascular response recorded under control conditions**
Individual data points illustrate the percent increases in measures of cerebrovascular flow, recorded in response to hypoxia and reported in the selected studies. Open symbols indicate the mean (with 95% confidence intervals) hypoxia-induced increases in measures of cerebrovascular flow observed under specific experimental conditions grouped into three categories to evaluate the effects of (*A*) the species of experimental subjects, (*B*) the type of anaesthesia and (*C*) the method used to measure the cerebrovascular response to hypoxia.

*Adenosine.* Eleven studies that investigated the role of adenosine-mediated signaling were selected for the analysis, including 9 experimental animal studies and 2 studies conducted in humans. Blockade of adenosine-mediated mechanisms was found to have the most significant effect on the cerebrovascular response to hypoxia [reduction by 49% (95% CIs: −64%, −34%), average of 9 experimental animal studies; $P < 0.001$] (Fig. 3). To block adenosine-mediated signaling, the following experimental treatments were used in animal studies: administration of theophylline (non-selective adenosine receptor antagonist) in rats (2 studies) and piglets (2 studies), administration of ZM-241385 (adenosine A2A/A2B receptor antagonist) in mice (1 study), genetic deletion of A2A receptor in mice (1 study), administration of adenosine deaminase in rats (1 study), administration of MRS1754 (A2A receptor antagonist) in rats (1 study) and administration of SCH58261 (A2A receptor antagonist) also in rats (1 study). All studies conducted in experimental animals reported reductions of hypoxia-induced responses in conditions of adenosine receptor blockade (Laudignon et al., 1990; Liu et al., 2015; McPhee & Maxwell, 1987; Miekisiak et al., 2008; Morii et al., 1987; Pelligrino et al., 1995; Simpson & Phillis, 1991). Theophylline reduced the hypoxic cerebrovascular response by 22% in one human study (Hoiland et al., 2017); aminophylline (theophylline/ethylenediamine combination) potentiated the cerebrovascular response

to hypoxia by 46% in another study conducted in human subjects (Bowton et al., 1988) (Fig. 3).

$K_{ATP}$ *channels.* Blockade of $K_{ATP}$ channels was found to have a significant effect on the expression of the cerebrovascular response to hypoxia (Fig. 4). Five studies selected for the analysis used the $K_{ATP}$ channel blocker glibenclamide in experiments conducted in rats (2 studies), piglets (1 study), rabbits (1 study) and humans (1 study). Data from 4 animal studies showed that treatment with glibenclamide reduced the cerebrovascular response to hypoxia by 37% (95% CIs: −52%, −22%, $P = 0.016$) on average (Liu et al., 2015; Shankar & Armstead, 1995; Taguchi et al., 1994; Tomiyama et al., 1999). In one study conducted in humans, glibenclamide was found to reduce the magnitude of the cerebrovascular response to hypoxia by 57% (Rocha et al., 2020) (Fig. 4).

*Nitric oxide.* A significant number of published studies investigated the role of a potent vasodilator NO in the mechanisms underlying the cerebrovascular response to hypoxia, but only 9 studies met the selection criteria for inclusion (Fig. 5). Five studies involved administration of a non-selective nitric oxide synthase (NOS) inhibitor $N^{\omega}$-nitro-L-arginine methyl ester (L-NAME) in dogs (1 study), piglets (1 study) and rats (3 studies), one study used an inhibitor of the neuronal and endothelial isoforms of NOS, $N$(G)-nitro-L-arginine (L-NNA) in piglets, one study used the NOS inhibitor 7-nitroindazole

| Intervention/study | Effect of intervention on the baseline (%) | Effect of intervention on the hypoxic response (%) | n (Control) | n (Experimental) | Species | |
|---|---|---|---|---|---|---|
| **Theophylline** | | | | | | |
| McPhee & Maxwell 1987 | 5.5 | −8.1 | 8 | 8 | Piglets | |
| Morii *et al.* 1987 | 7.9 | −73.0 | 14 | 15 | Rats | |
| Laudignon *et al.* 1990 | 11.1 | −51.8 | 6 | 5 | Piglets | |
| Pelligrino *et al.* 1995 | — | −46.8 | 5 | 5 | Rats | |
| **ZM-241385** | | | | | | |
| Miekisiak *et al.* 2008 | — | −39.5 | 13 | 12 | Mice | |
| **A2A receptor knockout** | | | | | | |
| Miekisiak *et al.* 2008 | — | −74.4 | 7 | 10 | Mice | |
| **Adenosine deaminase** | | | | | | |
| Simpson & Phillis 1991 | — | −44.3 | 6 | 6 | Rats | |
| **MRS1754** | | | | | | |
| Liu *et al.* 2015 | — | −28.3 | 20 | 13 | Rats | |
| **SCH58261** | | | | | | |
| Liu *et al.* 2015 | — | −75.3 | 20 | 18 | Rats | |
| **Mean (95% CI)** | | **−49.1 (−63.9, −34.2)** | | | | |
| **Aminophylline** | | | | | | |
| Bowton *et al.* 1988 | −31.7 | 45.9 | 5 | 5 | Humans | |
| **Theophylline** | | | | | | |
| Hoiland *et al.* 2017 | −8.7 | −22.3 | 8 | 8 | Humans | |

Percent increase in CBV response

**Figure 3. Effect of blocking adenosine-mediated signaling on the expression of the cerebrovascular (CBV) response to hypoxia**
Individual data points illustrate the percent change in the magnitude of the hypoxia-induced cerebrovascular response in conditions of pharmacological or genetic blockade of adenosine-mediated signaling and reported in the publications referenced. The diamond symbol illustrates the mean (with 95% confidence intervals) effect.

(7-NINA) in piglets, and one study used the NOS inhibitor N5-(1-iminoethyl)-L-ornithine (L-NIO) also in piglets. L-NAME was reported to either decrease, have no effect or increase the cerebrovascular response to hypoxia (Audibert et al., 1995; Christie et al., 2023; Kutzsche et al., 2002; Pelligrino et al., 1993, 1995). Treatments involving the use of L-NNA, 7-NINA or L-NIO were reported to reduce the responses to hypoxia (range from −8% to −53%; Armstead, 1999; Wilderman & Armstead, 1997, 1998). Analysis of the data reported in 8 experimental animal studies showed that blockade of NOS activity had no significant effect on the cerebrovascular response to hypoxia [decrease on average by 16% (95% CIs: −72%, 40%), $P = 0.48$]. In one study conducted in human subjects that met the selection criteria, administration of the non-selective NOS inhibitor $N$(G)-monomethyl-L-arginine (L-NMMA) reduced the cerebrovascular response to hypoxia by 24% (Hoiland et al., 2023) (Fig. 5).

*Products of arachidonic acid metabolism.* Several studies selected for the analysis investigated the potential role of arachidonic acid derivatives as mediators of the cerebrovascular response to hypoxia (Fig. 6). The experimental subjects used in these studies were humans, piglets and rats. The effects of the cyclooxygenase inhibitor indomethacin were tested in humans (2 studies) and piglets (2 studies). Studies conducted in rats used 14,15-epoxyeicosatrienoic acid (14,15-EEZE) to block the actions of 14,15-epoxyeicosatrienoic acid (1 study) and N-(methylsulfonyl)-2-(2-propynyloxy)-benzenehexanamide (MS-PPOH) to inhibit the enzymatic activity of epoxygenase (1 study). Analysis of the data from four animal studies showed that blockade of signaling pathways mediated by arachidonic acid derivatives had no significant effect on the cerebrovascular response to hypoxia [decrease by 34% (95% CIs: −79%, 11%), $P = 0.23$]. One study involving human participants reported that indomethacin reduced the magnitude

| Intervention/study | Effect of intervention on the baseline (%) | Effect of intervention on the hypoxic response (%) | n (Control) | n (Experimental) | Species |
|---|---|---|---|---|---|
| **Glibenclamide** | | | | | |
| Taguchi *et al.* 1994 | — | −43.9 | 8 | 8 | Rabbits |
| Shankar & Armstead 1995 | — | −52.5 | 5 | 5 | Piglets |
| Tomiyama *et al.* 1999 | −4.9 | −17.5 | 4 | 10 | Rats |
| Liu *et al.* 2015 | − | −33.3 | 20 | 13 | Rats |
| **Mean (95% CI)** | | **−36.8 (−51.6, −22.0)** | | | |
| **Glibenclamide** | | | | | |
| Rocha *et al.* 2020 | 14.1 | −56.6 | 9 | 9 | Humans |

**Figure 4. Effect of blocking ATP-sensitive potassium (K$_{ATP}$) channels on the expression of the cerebrovascular (CBV) response to hypoxia**
Individual data points illustrate the percent reduction in the magnitude of the hypoxia-induced cerebrovascular response in conditions of pharmacological blockade of K$_{ATP}$ channels reported in the publications referenced. The diamond symbol illustrates the mean (with 95% confidence intervals) effect.

| Intervention/study | Effect of intervention on the baseline (%) | Effect of intervention on the hypoxic response (%) | n (Control) | n (Experimental) | Species |
|---|---|---|---|---|---|
| **L-NAME** | | | | | |
| Pelligrino *et al.* 1993 | −52.9 | 156.2 | 4 | 5 | Rats |
| Audibert *et al.* 1995 | 11.4 | −93.7 | 8 | 8 | Dogs |
| Pelligrino *et al.* 1995 | — | −2.6 | 6 | 3 | Rats |
| Kutzsche *et al.* 2002 | 32.5 | −100.0 | 8 | 8 | Piglets |
| Christie *et al.* 2023 | — | 11.0 | 6 | 7 | Rats |
| **L-NNA** | | | | | |
| Armstead 1999 | — | −35.7 | 8 | 8 | Piglets |
| **7-NINA** | | | | | |
| Wilderman & Armstead 1997 | — | −52.9 | 7 | 7 | Piglets |
| **L-NIO** | | | | | |
| Wilderman & Armstead 1998 | — | −8.0 | 6 | 6 | Piglets |
| **Mean (95% CI)** | | **−15.7 (−71.5, 40.1)** | | | |
| **L-NMMA** | | | | | |
| Hoiland *et al.* 2023 | −1.4 | −24.0 | 11 | 11 | Humans |

**Figure 5. Effect of blocking nitric oxide-mediated signaling on the expression of the cerebrovascular (CBV) response to hypoxia**
Individual data points illustrate the percent change in the magnitude of the hypoxia-induced cerebrovascular response in conditions of pharmacological blockade of nitric oxide synthase activity and reported in the publications referenced. The diamond symbol illustrates the mean (with 95% confidence intervals) effect. 7-NINA, 7-nitroindazole; L-NAME, $N$(G)-nitro-L-arginine methyl ester; L-NIO, $N5$-(1-iminoethyl)-L-ornithine; L-NMMA, $N$(G)-monomethyl-L-arginine; L-NNA, $N$(G)-nitro-L-arginine.

of the cerebrovascular response to hypoxia by 14% (Kellawan et al., 2020). In another study, indomethacin was found to potentiate the response to hypoxia by 17% (Fan et al., 2011) (Fig. 6). It was also reported that blockade of 20-hydroxyeicosatetraenoic acid (20-HETE) synthesis using HET0016 decreased the magnitude of the hypoxia-induced cerebrovascular response in rats by 10% (Liu et al., 2015) (Fig. 6).

*Other signaling mechanisms.* Studies designed to investigate the role of calcium-sensitive potassium channels ($K_{Ca}$) were performed in experimental animals, specifically in piglets. The selected studies examined the effects of $K_{Ca}$ channel blockers iberiotoxin (2 studies) and paxilline (1 study). Decreases (by 51% and 43%) of the cerebrovascular response to hypoxia were observed following administration of iberiotoxin (Armstead, 1998; Ben-Haim & Armstead, 2000). Paxilline, however, had no effect on the hypoxia-induced cerebrovascular response (Kanu & Leffler, 2007) (Fig. 7). As there were only three studies investigating the role of $K_{Ca}$ channels, a meta-analysis was not performed.

Three experimental animal studies explored the potential role of signaling by catecholamines. The studies included in this analysis examined the effects of the dopamine receptor antagonist *N*-methylchlorproma-zine in rabbits (1 study), $\alpha$-adrenoceptor blocker phenoxybenzamine in rabbits (1 study) and $\alpha_1$-adrenoceptor inhibitor prazosin in sheep (1 study). The data are presented in Fig. 7. Considering the diverse pharmacological properties of the agents used in these studies, a meta-analysis of the data was not performed.

Significant reductions of the hypoxia-induced cerebrovascular response were observed in one study which tested the effects of several experimental genetic strategies designed to block signaling mechanisms mediated by hydrogen sulphide. In mice, genetic knockout of heme oxygenase 2, cystathionine $\beta$-synthase (CBS) or cystathionine $\gamma$-lyase (CSE) reduced the cerebrovascular responses to hypoxia by 61%, 99% and 18%, respectively (Morikawa et al., 2012) (Fig. 7).

Three experimental animal studies targeted histamine receptors with D-chlorpheniramine, roxatidine and famotidine. Treatment with D-chlorpheniramine ($H_1$ receptor blocker) led to a 37% reduction of the cerebrovascular response to hypoxia. Treatment with $H_2$ receptor blockers, roxatidine and famotidine, caused 96% and 99% reductions of the response respectively (Audibert et al., 1991, 1998) (Fig. 7).

Two studies targeted the glutamatergic mechanisms in the experiments conducted in rats. Blockade of metabotropic glutamate receptors with 2-methyl-6-(phenylethynyl)pyridine (MPEP) and LY367385 or NMDA receptors with MK-801 reduced the cerebrovascular responses to hypoxia by 84% and 44%, respectively (Liu et al., 2015; Pelligrino et al., 1995) (Fig. 7). Two studies targeted sodium channels and reported 100% (Pelligrino et al., 1995) and 48% reduction (Wilderman & Armstead, 1997) of the hypoxic vasodilatory response following the application of tetrodotoxin (TTX).

Two individual studies targeted other specific mechanisms and reported reductions of the hypoxia-induced cerebrovascular responses in the experiments involving treatments with $\beta$-funaltrexamine

| Intervention/study | Effect of intervention on the baseline (%) | Effect of intervention on the hypoxic response (%) | n (Control) | n (Experimental) | Species |
|---|---|---|---|---|---|
| **Indomethacin** | | | | | |
| Coyle *et al.* 1993 | −31.0 | −32.8 | 9 | 8 | Piglets |
| Coyle *et al.* 1995 | −31.3 | 30.1 | 9 | 8 | Piglets |
| **14,15-EEZE** | | | | | |
| Liu *et al.* 2015 | − | −74.1 | 20 | 13 | Rats |
| **MS-PPOH** | | | | | |
| Liu *et al.* 2015 | − | −59.1 | 20 | 13 | Rats |
| **Mean (95% CI)** | | −34.0 (−79.1, 11.1) | | | |
| **HET0016** | | | | | |
| Liu *et al.* 2015 | − | −10.4 | 20 | 15 | Rats |
| **Indomethacin** | | | | | |
| Fan *et al.* 2011 | −23.8 | 17.1 | 12 | 12 | Humans |
| Kellawan *et al.* 2020 | −41.1 | −14.1 | 9 | 9 | Humans |

Percent change in CBV response
−100  −50  0  50  100

**Figure 6. Effects of blocking signaling pathways mediated by products of arachidonic acid metabolism on the expression of the cerebrovascular (CBV) response to hypoxia**
Individual data points illustrate the percent change in the magnitude of the hypoxia-induced cerebrovascular response in conditions of pharmacological blockade of cyclooxygenase (indomethacin), epoxygenase (MS-PPOH), the actions of 14,15-epoxyeicosatrienoic acid (14,15-EEZE) and the synthesis of 20-hydroxyeicosatetraenoic acid (HET0016). The diamond symbol illustrates the mean (with 95% confidence intervals) effect.

to inhibit opioid signaling (reduction by 31%) (Armstead, 1995) (Fig. 7), and Rp-8-Br-cAMPS to inhibit cAMP-mediated signaling (reduction by 30%) (Ben-Haim & Armstead, 2000) (Fig. 7).

**Risk of bias.** The summary results of the risk of bias analysis are presented in Fig. 8.

*Human studies.* It was found that 50% (3 publications), 16.6% (1 publication) and 33.3% (2 publications) of selected articles describing the results of studies conducted in human participants had 'low', 'unclear' and 'high' risk of bias, respectively. Four publications reported random sequence generation (Fan et al., 2011; Hoiland et al., 2017, 2023; Kellawan et al., 2020). Three publications reported blinding of outcome assessment

(Hoiland et al., 2017, 2023; Kellawan et al., 2020). The bias due to selective reporting and reporting of incomplete outcomes was unclear for all the selected studies. All articles indicated that the studies complied with human welfare regulations and four publications provided statements of the potential conflict of interest (Hoiland et al., 2017, 2023; Kellawan et al., 2020; Rocha et al., 2020).

*Experimental animal studies.* It was found that 11% (3 publications) and 89% (25 publications) of the experimental animal studies that met the inclusion criteria had 'unclear' and 'high' risk of bias, respectively. Eighteen studies showed low risk of bias due to random sequence generation (Audibert et al., 1991, 1995, 1998; Christie et al., 2023; Coyle et al., 1993, 1995; Laudignon et al., 1990; Liu et al., 2015; McPhee & Maxwell, 1987;

| | Intervention/study | Effect of intervention on the baseline (%) | Effect of intervention on the hypoxic response (%) | n (Control) | n (Experimental) | Species |
|---|---|---|---|---|---|---|
| Catecholamines | **N-Methylchlorpromazine** Weiss & Buchweitz-Milton 1988 | −3.5 | −14.4 | 7 | 7 | Rabbits |
| | **Phenoxybenzamine** Weiss & Buchweitz-Milton 1988 | −3.5 | −40.9 | 7 | 7 | Rabbits |
| | **Prazosin** Wagerle *et al.* 1983 | 10.6 | −37.1 | 5 | 6 | Sheep |
| cAMP | **Rp-8-Br-cAMPS** Ben Haim & Armstead 2000 | − | −30.0 | 7 | 7 | Piglets |
| Glutamate | **MPEP + LY367385** Liu *et al.* 2015 | − | −83.6 | 20 | 15 | Rats |
| | **MK-801** Pelligrino *et al.* 1995 | − | −43.7 | 5 | 5 | Rats |
| Histamine | **Roxatidine** Audibert *et al.* 1991 | 12.4 | −96.5 | 6 | 6 | Dogs |
| | **Famotidine** Audibert *et al.* 1991 | 10.1 | −99.0 | 8 | 8 | Dogs |
| | **d-Chlorpheniramine** Audibert *et al.* 1998 | −2.0 | −37.0 | 8 | 8 | Dogs |
| Hydrogen sulfide | **Haem oxygenase 2 knockout** Morikawa *et al.* 2012 | − | −60.6 | 12 | 10 | Mice |
| | **Cystathionine β-synthase knockout** Morikawa *et al.* 2012 | − | −98.7 | 12 | 3 | Mice |
| | **Cystathionine γ-lyase knockout** Morikawa *et al.* 2012 | − | −18.4 | 12 | 6 | Mice |
| K$_{Ca}$ channels | **Iberiotoxin** Armstead 1998 Ben Haim & Armstead 2000 | − − | −50.7 −43.2 | 5 7 | 5 7 | Piglets Piglets |
| | **Paxilline** Kanu and Leffler 2007 | −6.9 | −4.8 | 8 | 8 | Piglets |
| Opioids | **β-funaltrexamine** Armstead 1995 | − | −31.4 | 5 | 5 | Piglets |
| VGSCs | **Tetrodotoxin** Pelligrino *et al.* 1995 Wilderman & Armstead 1997 | − − | −100.0 −47.9 | 5 7 | 5 7 | Rats Piglets |

−100  −50  0  50  100
**Percent change in CBV response**

**Figure 7. Effects of blocking signaling pathways involving catecholamines, cAMP, glutamate, histamine, hydrogen sulphide, calcium-activated potassium (K$_{Ca}$) channels, opioids and voltage-gated sodium channels (VGSCs) on the expression of the cerebrovascular (CBV) response to hypoxia**
Individual data points illustrate the percent change in the magnitude of the hypoxia-induced cerebrovascular response in conditions of pharmacological or genetic blockade of hypothesized signaling pathways.

*A*

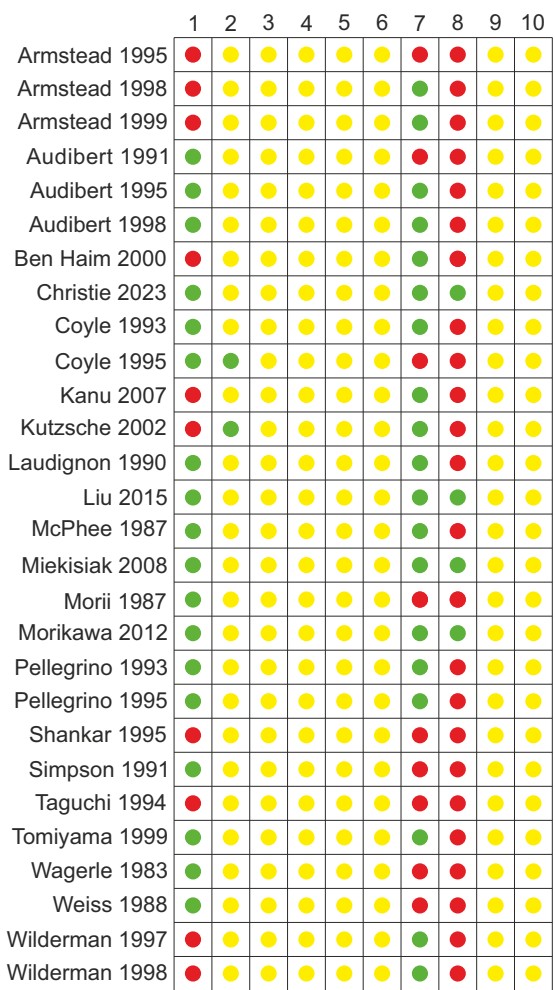

*B*

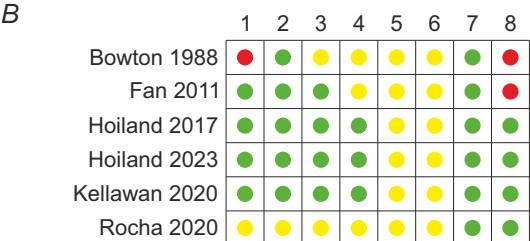

**Figure 8. Risk of bias assessment of the included studies using the Cochrane and SYRCLE risk of bias tools for human and animal studies, respectively**

Green – low risk; yellow – unclear risk (insufficient data); red – high risk. *A*: 1, random sequence generation (selection bias); 2, allocation concealment (selection bias); 3, blinding of experimenters (performance bias); 4, blinding of outcome assessment (detection bias); 5, incomplete outcome data (attrition bias); 6, selective reporting (reporting bias); 7, statement of compliance with animal welfare regulations; 8, statement of potential conflict of interest; 9, random outcome assessment; 10, random housing. *B*: 1, random sequence generation (selection bias); 2, allocation concealment (selection bias); 3, blinding of experimenters (performance bias); 4, blinding of outcome assessment (detection bias); 5, incomplete outcome data (attrition bias); 6, selective reporting (reporting bias); 7, statement of compliance with human welfare regulations; 8, statement of potential conflict of interest.

Miekisiak et al., 2008; Morii et al., 1987; Morikawa et al., 2012; Pelligrino et al., 1993, 1995; Tomiyama et al., 1999; Wagerle et al., 1983; Weiss & Buchweitz-Milton, 1988), and two studies reported allocation concealment (Coyle et al., 1995; Kutzsche et al., 2002). All studies were classified as having an unclear risk of bias in the dimensions of blinding of personnel, blinding of outcome assessment, incomplete outcome data, selective reporting, random outcome assessment and random housing. Eighteen studies reported compliance with animal welfare regulations (Armstead, 1998, 1999; Audibert et al., 1995, 1998; Ben-Haim & Armstead, 2000; Christie et al., 2023; Coyle et al., 1993, 1995; Kanu & Leffler, 2007; Kutzsche et al., 2002; Liu et al., 2015; Miekisiak et al., 2008; Morikawa et al., 2012; Pelligrino et al., 1993, 1995; Tomiyama et al., 1999; Wilderman & Armstead, 1997, 1998). Four publications provided statements of potential conflict of interest (Christie et al., 2023; Liu et al., 2015; Miekisiak et al., 2008; Morikawa et al., 2012).

## Discussion

Here we describe the results of a systematic review of published data obtained in experimental animal and human studies that investigated the mechanisms of hypoxic cerebral vasodilation – a fundamental adaptive response underlying increases in CBF during hypoxia. Our primary outcome measure was the percent reduction of the cerebrovascular response to hypoxia in conditions of either pharmacological or genetic blockade of hypothesized mechanisms suggested by studies conducted in experimental animals or in humans. By surveying the entire literature, we aimed to identify the key signaling mechanism(s) responsible for regulation of CBF in conditions of reduced oxygen supply to the brain. Figure 9 illustrates the key hypothesized pathways suggested by studies included in this analysis.

Database searches returned 4848 publications. Only 246 articles met the selection criteria for full-text screening; from these, 28 research articles describing the results of experimental animal studies and 6 articles describing studies conducted in humans met the criteria for inclusion in the analysis. First, we evaluated the hypoxic cerebrovascular responses recorded under control conditions in all the selected studies. Significant heterogeneity in the reported results was found; the magnitude of the hypoxic cerebrovascular response recorded under control conditions varied between 17% (study in humans; Bowton et al., 1988) and 196% (study in rats; Morii et al., 1987). The expression of the hypoxic cerebrovascular response was found to be dependent on the species of experimental subjects, the method used to measure the cerebrovascular response to hypoxia, and also on whether the experimental subjects were conscious or anaesthetized, and if anaesthetized, on the type of

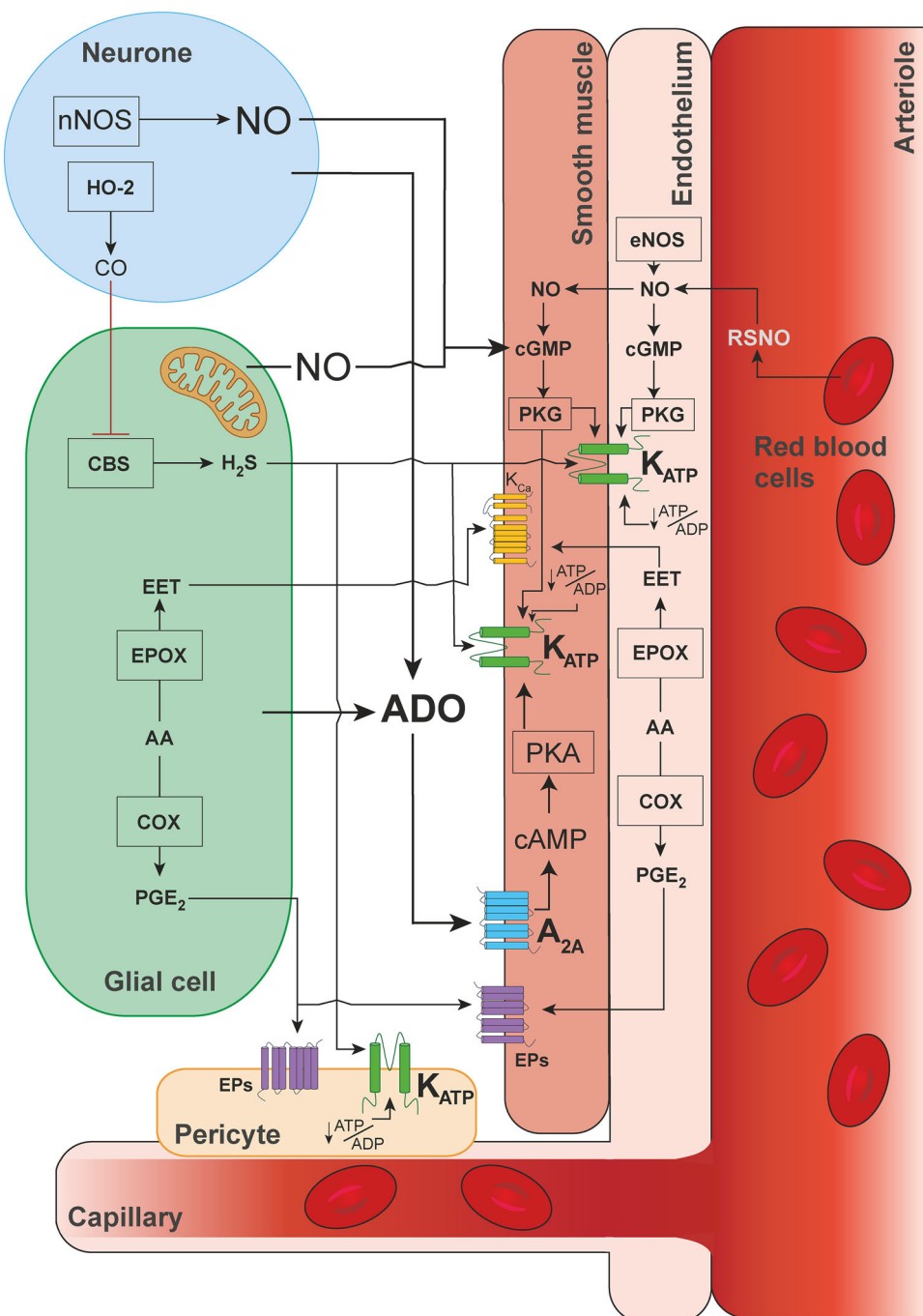

**Figure 9. Hypothesized signaling mechanisms of hypoxic cerebral vasodilation**
Schematic illustration of key hypothesized pathways mediating the cerebrovascular response to hypoxia suggested by the results of studies included in this analysis. $A_{2A}$, adenosine receptor 2A; AA, arachidonic acid; ADO, adenosine; ADP, adenosine diphosphate; ATP, adenosine triphosphate; cAMP, cyclic adenosine monophosphate; CBS, cystathionine beta-synthase; cGMP, cyclic guanosine monophosphate; CO, carbon monoxide; COX, cyclo-oxygenase; EET, epoxyeicosatrienoic acid; eNOS, endothelial nitric oxide synthase; EPOX, cytochrome P450 epoxygenase; EPs, prostaglandin receptors; $H_2S$, hydrogen sulphide; HO-2, heme oxygenase 2; $K_{ATP}$, ATP-sensitive potassium channels; nNOS, neuronal nitric oxide synthase; NO, nitric oxide; $PGE_2$, prostaglandin E2; PKA, protein kinase A; PKG, protein kinase G; RSNO, *S*-nitrosothiol.

the anaesthetic agent used. In all the studies included in this analysis, isocapnic conditions were maintained throughout the experimental protocols. This is important as hyperventilation in response to systemic hypoxia leads to hypocapnia [low partial pressure of arterial $CO_2$ ($P_{aCO_2}$)] and alkalosis, promoting cerebrovascular constrictions, which would be expected to oppose and reduce the effect of hypoxia on cerebral vasculature (Willie et al., 2014).

The expression of the cerebrovascular response to hypoxia is obviously dependent on the strength of the hypoxic stimulus. Ideally, the comparisons of the responses between studies should take into account the measures of cerebrovascular reactivity to hypoxia which would require analysis of the response slope ($\Delta CBF/\Delta P_{O_2}$). However, the data reporting the slopes of the responses or data that would be required to calculate the slopes were not available for the majority of the experimental animal studies. Many publications just reported that hypoxia was induced by reduction in the level of inspired oxygen, but the effects of the experimental stimuli on $S_{aO_2}$ and/or $P_{aO_2}$ were often not reported. Therefore, the lack of between-study comparisons of the responses and the effects of treatments normalized to the stimulus strength is a significant limitation of this analysis.

All human studies included in the analysis were performed without the use of anaesthesia at clamped $P_{aCO_2}$ conditions. Compared to the responses recorded in the majority of the experimental animal studies, the hypoxia-induced increases in cerebrovascular flow in humans were smaller (increases on average by 27±3%), but the least variable, with the magnitude of the control hypoxia-induced responses reported within the range between 17% (Bowton et al., 1988) and 35% (Hoiland et al. 2023). The differences in the magnitude of the hypoxia-induced cerebrovascular response recorded in humans *versus* the majority of experimental animal studies could potentially be explained by more severe levels of hypoxia that are often applied in animal studies.

Despite significant heterogeneity in the reported data, the results of our analysis point to one signaling pathway which is likely to play an important role in regulation of CBF during hypoxia. Of all the experimental conditions, blockade of adenosine-mediated signaling was found to have the most significant effect in reducing the cerebrovascular response to hypoxia (inhibition by 49%; an average of 9 experimental animal studies). Hypoxia-induced cerebrovascular responses were also found to be significantly reduced in conditions of $K_{ATP}$ channel blockade (inhibition by 37%; average of 4 experimental animal studies). Considering that $K_{ATP}$ channels in vascular smooth muscle cells, endothelial cells and capillary pericytes are ultimately responsible for the effects of adenosine on the vasculature (Aziz et al., 2017; Kleppisch & Nelson, 1995; Sancho et al., 2022;

Tinker et al., 2018), this analysis suggests that one of the key mechanisms mediating the hypoxic cerebral vasodilation (accounting for ∼50% of the response) involves the release of adenosine and modulation of the activity of vascular $K_{ATP}$ channels. Indeed, there is significant evidence that brain hypoxia is associated with the release of adenosine (Dale et al., 2000; Frenguelli et al., 2003) and that adenosine is acting through A2A receptors and the cAMP/protein kinase A pathway to activate vascular $K_{ATP}$ channels leading to hyperpolarization of the endothelial and smooth muscle cells causing vasodilation (Sancho et al., 2022). There is also evidence obtained in animal models suggesting that $K_{ATP}$ channels are critically important for the maintenance of resting CBF and brain parenchymal $P_{O_2}$ (Hosford et al., 2019). However, two studies performed in humans did not show consistent effects of the adenosine receptor antagonist theophylline on the expression of cerebrovascular response to hypoxia (Bowton et al., 1988; Hoiland et al., 2017). This outcome of human studies is surprising considering that all studies conducted in experimental animals reported reductions of the response when adenosine receptors were blocked pharmacologically or deleted genetically. It is unknown, however, whether sufficient blockade of cerebrovascular adenosine receptors was achieved following administration of theophylline in human studies. If brain adenosine receptors were effectively blocked by theophylline, then the lack of an effect of this treatment on hypoxic cerebral vasodilation in humans can potentially be explained by the level of hypoxia applied in these studies not being sufficient to trigger the release of adenosine by the brain tissue. However, in a separate human study, blockade of $K_{ATP}$ channels was found to reduce the cerebrovascular response to hypoxia by 57% (Rocha et al., 2020), the outcome which is fully consistent with the data obtained in experimental animals.

In the context of this discussion, it is important to consider that up to 90% of the world's adult population regularly consume the adenosine receptor antagonist caffeine. The amounts of caffeine in drinks and foods are sufficient to raise plasma concentrations and inhibit adenosine receptors in the brain (Fredholm et al., 1999). Although most of the human studies included in this analysis required the participants to abstain from caffeine consumption for several hours immediately prior to the experimental testing, the impact of life-long caffeine consumption on the expression of the hypoxic cerebrovascular response is unknown.

Our analysis also suggests that in addition to adenosine-mediated signaling, the other vasodilatory signaling mechanisms are recruited, either in parallel or sequentially, and are responsible for ∼50% of the hypoxia-induced cerebrovascular response. One of the other notable mechanisms involves signaling mediated

by NO. Several experimental animal and human studies have examined the effects of pharmacological blockade of NOS activity; all studies included in this review used non-selective inhibitors that block the activities of two or all three NOS isoforms. Analysis of data from the experimental animal studies showed that blockade of NOS had little effect on the cerebrovascular response to hypoxia (inhibition by 16%; average of 8 experimental animal studies). It is important to note that inhibition of NOS is often associated with significant (by ∼30%) reductions of baseline CBF (discussed in Hosford & Gourine, 2019), potentially confounding the measurements taken when hypoxia is applied in conditions of systemic NOS blockade. However, changes in the baseline flow following NOS blockade were not reported in the majority of the selected studies. Nevertheless, considering that almost complete blockade of brain NOS activity can be achieved using pharmacological tools in the experiments of this type (Zhang et al., 1995), the results of this analysis suggest that NO produced by NOS plays a minor role, if any, in mediating the cerebrovascular response to hypoxia.

A recent human study showed that the NOS inhibitor L-NMMA reduced the cerebrovascular response to hypoxia by 24% (Hoiland et al., 2023) and provided other evidence suggesting that NO-mediated signaling may contribute to hypoxic cerebral vasodilation. The study reported that in humans the higher CBF response to hypoxia was associated with greater trans-cerebral release of $S$-nitrosothiol and that the magnitude of the cerebrovascular response to hypoxia was inversely correlated to the arterial concentration of haemoglobin (Hoiland et al., 2023) – an effective scavenger of NO (Eich et al., 1996). Alternative mechanisms of NO generation, which do not involve NO production by NOS and are therefore not sensitive to NOS blockade, have been proposed. These mechanisms include NO production through the reduction of nitrite anion ($NO_2^-$), an alternative (NOS independent) mechanism of NO generation requiring an electron donor and haem- or molybdenum cofactor-containing protein (Christie et al., 2023; DeMartino et al., 2019), and the release of NO from $S$-nitrosothiols by red blood cells following the allosteric shift upon deoxygenation (Hoiland et al., 2023). In the Hoiland et al. (2023) study, L-NMMA treatment led to a 24% reduction of the cerebrovascular response to hypoxia and a 23% reduction in arterial concentration of $NO_2^-$ and $S$-nitrosothiol, indicating that systemic inhibition of NOS lowers the amount of nitrite available for NO production via a reductive pathway. It is important to emphasize that although the cellular and molecular mechanisms underlying cerebrovascular oxygen sensing are not well understood, studies have demonstrated increased production and release of both adenosine and NO during brain tissue hypoxia (Christie et al., 2023; Dale et al., 2000; Frenguelli et al., 2003).

Several studies have investigated the potential role of arachidonic acid derivatives as mediators of the cerebrovascular response to hypoxia. The data from four animal studies showed that the blockade of signaling pathways mediated by eicosanoids reduces the cerebrovascular response to hypoxia. In particular, blockade of signaling mediated by epoxygenase products (epoxyeicosatrienoic acids) appeared to have an effect in reducing the hypoxia-induced cerebrovascular dilations. Two studies conducted in humans involving systemic blockade of cyclooxygenase activity with indomethacin reported conflicting results (Fan et al., 2011; Kellawan et al., 2020). That indomethacin markedly reduces the cerebrovascular reactivity to $CO_2$, as reported by many animal and human studies (Hosford et al., 2022), indicates the efficacy of indomethacin action on brain vasculature. It is important to note that indomethacin used in these studies as a non-specific cyclooxygenase inhibitor is also a potent inhibitor of cyclic AMP-dependant protein kinase activity (Goueli & Ahmed, 1980), which is integral to any vasomotor response. Therefore, the results of the experiments that used indomethacin as a cyclooxygenase inhibitor should be interpreted with caution.

This survey of the literature also identified a number of experimental animal studies that investigated the potential involvement of several other signaling mechanisms. The relevant studies that met the inclusion criteria described the results of the experiments involving blockade of mechanisms involving calcium-sensitive potassium channels, cAMP, catecholamines, glutamate, histamine, opioids and sodium channels. Various degree reductions of the cerebrovascular response to hypoxia were reported. Results obtained in one notable study conducted in mice suggested that hydrogen sulphide produced by the astroglial enzyme cystathionine $\beta$-synthase may act as a mediator of hypoxic cerebral vasodilation (Morikawa et al., 2012) (of note, the vasodilatory action of $H_2S$ is mediated by $K_{ATP}$ channels; Mustafa et al., 2011). However, no follow-up studies have addressed the role of $H_2S$-dependent mechanisms further. As there are very few studies investigating these individual pathways (with only one relevant study found for three of these pathways), definitive conclusions about their importance cannot be drawn at present. Nonetheless, our analysis provides insight into the potential significance of these mechanisms in mediating the effects of hypoxia on CBF and emphasizes the necessity for further research in this area.

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

## Additional information

### Competing interests

None declared.

### Author contributions

A.M. carried out database searches under the guidance of D.M. and N.M.; A.M., A.B., S.M.M., S.N., S.M.T. and Q.A. analysed data. N.M. and A.V.G. wrote the draft of the manuscript. All authors contributed to writing the paper and revised the article critically for important intellectual content. All authors approved the final version of the manuscript. All persons designated as authors qualify for authorship, and all those who qualify for authorship are listed.

### Funding

This work was supported by The Wellcome Trust (A.V.G.: refs. 200893/Z/16/Z and 223057/Z/21/Z) and British Heart Foundation (Ref. RG/19/5/34463).

### Keywords

adenosine, brain, cerebral blood flow, hypoxia, nitric oxide, oxygen, vasodilation

### Supporting information

Additional supporting information can be found online in the Supporting Information section at the end of the HTML view of the article. Supporting information files available:

**Peer Review History**
**Table S1-S5**

