## [Peer Review History · The Journal of Physiology]

On the mechanisms of brain blood flow regulation during hypoxia

Alexander Mascarenhas, Alice Braga, Sara Maria Majernikova, Shereen Nizari, Debora Marletta, Shefееq M Theparambil, Qadeer Aziz, Nephtali Marina, and Alexander V Gourine

DOI: 10.1113/JP285060

Corresponding author(s): Alexander Gourine (a.gourine@ucl.ac.uk)

The following individual(s) involved in review of this submission have agreed to reveal their identity: Ryan L Hoiland (Referee #2)

Review Timeline:

Submission Date:	25-Aug-2023
Editorial Decision:	19-Sep-2023
Revision Received:	22-Mar-2024
Editorial Decision:	05-Apr-2024
Revision Received:	13-May-2024
Accepted:	20-May-2024

Senior Editor: Laura Bennet

Reviewing Editor: Daniel Zoccal

Transaction Report:

Dear Dr Gourine,

Re: JP-TR-2023-285060 "Signalling mechanisms regulating cerebral blood flow during hypoxia" by Alexander Mascarenhas, Alice Braga, Shereen Nizari, Debora Marletta, Shefeeq M Theparambil, Nephtali Marina, and Alexander V Gourine

Thank you for submitting your manuscript to The Journal of Physiology. It has been assessed by a Reviewing Editor and by 2 expert referees and we are pleased to tell you that it is potentially acceptable for publication following satisfactory major revision.

REVISION CHECKLIST:

We look forward to receiving your revised submission.

Yours sincerely,

Professor Laura Bennet
Senior Editor
The Journal of Physiology
<https://jp.msubmit.net>
<http://jp.physoc.org>
The Physiological Society
Hodgkin Huxley House
30 Farringdon Lane
London, EC1R 3AW
UK
<http://www.physoc.org>
<http://journals.physoc.org>

REQUIRED ITEMS FOR REVISION

-Please include an Abstract Figure file, as well as the figure legend text within the main article file. The Abstract Figure is a piece of artwork designed to give readers an immediate understanding of the Review Article and should summarise the main conclusions. If possible, the image should be easily 'readable' from left to right or top to bottom. It should show the physiological relevance of the Review so readers can assess the importance and content of the article. Abstract Figures should not merely recapitulate other figures in the Review. Please try to keep the diagram as simple as possible and without superfluous information that may distract from the main conclusion of the Review. Abstract Figures must be provided by authors no later than the revised manuscript stage and should be uploaded as a separate file during online submission labelled as File Type 'Abstract Figure'. Please ensure that you include the figure legend in the main article file. All Abstract Figures will be sent to a professional illustrator for redrawing and you may be asked to approve the redrawn figure before your paper is accepted.

-Your MS must include a complete "Additional information section" with the following 4 headings and content:

Competing Interests: A statement regarding competing interests. If there are no competing interests, a statement to this effect must be included. All authors should disclose any conflict of interest in accordance with journal policy.

Author contributions: Each author should take responsibility for a particular section of the study and have contributed to writing the paper. Acquisition of funding, administrative support or the collection of data alone does not justify authorship; these contributions to the study should be listed in the Acknowledgements. Additional information such as 'X and Y have contributed equally to this work' may be added as a footnote on the title page.

It must be stated that all authors approved the final version of the manuscript and that all persons designated as authors qualify for authorship, and all those who qualify for authorship are listed.

Funding: Authors must indicate all sources of funding, including grant numbers. If authors have not received funding, this must be stated.

It is the responsibility of authors funded by RCUK to adhere to their policy regarding funding sources and underlying research material. The policy requires funding information to be included within the acknowledgement section of a paper. Guidance on how to acknowledge funding information is provided by the Research Information Network. The policy also requires all research papers, if applicable, to include a statement on how any underlying research materials, such as data, samples or models, can be accessed. However, the policy does not require that the data must be made open. If there are considered to be good or compelling reasons to protect access to the data, for example commercial confidentiality or legitimate sensitivities around data derived from potentially identifiable human participants, these should be included in the statement.

Acknowledgements: Acknowledgements should be the minimum consistent with courtesy. The wording of acknowledgements of scientific assistance or advice must have been seen and approved by the persons concerned. This section should not include details of funding.

-Author profile(s) must be uploaded via the submission form. Authors should submit a short biography (no more than 100 words for one author or 150 words in total for two authors) and a portrait photograph of the two leading authors on the paper. These should be uploaded, clearly labelled, with the manuscript submission. Any standard image format for the photograph is acceptable, but the resolution should be at least 300 dpi and preferably more. A group photograph of all authors is also acceptable, providing the biography for the whole group does not exceed 150 words.

EDITOR COMMENTS

Reviewing Editor:

This systemic review examined clinical and experimental studies that explored the mechanisms associated with cerebral blood flow regulation during hypoxia. Overall, the article is well-written and brings helpful information to the field. The review underwent evaluation by two experts who expressed enthusiasm and optimism about its content. Additionally, the referees provided relevant feedback for the authors to consider in relation to the analyzed data. These insights have the potential to enhance the review's impact and provide additional valuable information.

REFEREE COMMENTS

Referee #1:

This is a nice systematic review of the literature examining papers that have described the cell pathways involved in cerebral blood flow increases to acute hypoxia in mammals, including a small number of human studies. From an initial search of >5000 papers, the authors narrow it down to 34: 28 animal studies and 6 human studies. They find that the adenosine system, including activation of KATP channels shows the most consistent and largest block of hypoxia induced vasodilation, whereas other pathways are either not involved or more minorly contribute. The paper adheres to systematic review standards. The authors carefully track all the main variables that are different between the papers and clearly divulge them so that readers can make informed interpretations, including the type of animal used, the type of anesthesia, ventilation or not, how CBF was measured and other variables too. Importantly, they also track the confidence interval and report the average p values across papers. All papers included needed a p-values less than 0.05 on the primary measure (a percent reduction in hypoxic-induced vasodilation by the experimental manipulation). One minor short coming is that there are not very many papers (only 34), but this is still informative and useful. I have only a few minor points for the authors.

It may be worth while for the authors to consider the error in each of the averages taken from these papers. For example, one paper may show a 40% effect with a $p=0.05$, whereas another paper may show an effect of 10% with a $p=0.001$. Not considering the error from each paper produces a bias towards the average value itself and could affect the interpretation. Similarly, one can make an argument based on the sample size, where large sample sizes could be considered more influential. This could be done by taking a weighted average from each study, and/or calculating the average standard deviation across all the data sets from the original papers, and report that error instead of the SD from the averaging the average values from the papers. I am not saying the authors must do this, but they should run some tests to see if the outcome and interpretation changes.

The authors say: "one study used an inhibitor of the neuronal and endothelial isoforms of NOS - L-NNA in piglets, one study used a neuronal NOS inhibitor 7-NINA in piglets, and 13 one study used guanyl cyclase inhibitor L-NIO in piglets". As far as I understand, there are no highly selective antagonists for nNOS vs eNOS. 7-NINA is even stated to be a general NOS inhibitor by commercial vendors. Also, L-NIO is not a cGMP inhibitor, but a non-selective antagonist of NOS. Perhaps this sentence could be rephrased, stating that several different NOS inhibitors have been tested.

In the figures, everything is black and white and all the symbols are the same (circles) and it is difficult to follow which circles are connect to which variables/treatments. At least make each group a different symbol: square, triangle, filled unfilled etc., and each text label could also have that corresponding symbol beside the text.

Also, especially for fig3 and fig7, because the graph on the right is very tall, vertical dotted lines are needed for the x-axis ticks, such as 50, 100 etc. It is difficult to see where the points lay without this.

Referee #2:

I very much enjoyed reading this systematic review and meta-analysis. The mechanisms of hypoxic cerebral vasodilation is one of the areas of research I am most passionate about and I believe the author's have produced an important piece of literature for this field. I hope that the authors find my comments helpful. Sincerely, Ryan Hoiland.

Major Comment #1: One major comment is in regard to the data that is extracted and used to represent the change in CBF during hypoxia. For this I have the following comments/concerns:

1) Can the authors specify if they were comparing the %reduction in the absolute change in CBF (i.e., delta CBF), or the slope response of the change in CBF (i.e., delta CBF / delta SaO₂). It is unclear to me exactly what the extracted data was, although I am fairly certain the authors used the absolute change in CBF in their analyses (e.g., data in Figure 2). There are important implications for considering the variation in stimulus magnitude that can occur between experimental trials within a study, as well as between studies - without taking this into consideration, it is difficult to interpret the data. For example, it is unclear if the difference in CBF in the different subgroups that are reported in figure 2 are due to variation in stimulus magnitude. By reporting these data as the slope of the response (i.e., cerebrovascular reactivity to hypoxia) it would make direct comparisons easier and the data more informative. I urge the authors to consider this as a more appropriate outcome variable.

2) The fact that %changes in CBF are dependent upon the baseline blood flow, which could be influenced by any of the pharmacologic / genetic interventions included, also raises issues for interpreting the data. For example, the change in absolute flow response for the Bowton 1988 study is approximately +50% (8.5mL/100g/min vs. 12.4mL/100g/min), whereas the change in the %change in flow is inflated because of the lower baseline blood flow (the +113.9% you report in Figure 3). This is similar to the Pelligrino 1993 study used in the nitric oxide meta-analysis. Conversely, the 24% reduction reported in the Hoiland 2023 study, is based on the change in the absolute flow response, not the %change in flow. To calculate the %change in the CBF response following a drug based on both the %flow response and absolute flow response, with no indication for when one or the other is being used is confusing and hinders the level to which the data can be compared and subsequently interpreted. I urge the authors add the control CBF response (absolute CBF change scaled to a stimulus variable such as SaO₂ or CaO₂), the intervention CBF response, and the units of measure (e.g., mL/100g/min, cm/s, etc.) to the tables they include with their forest plots. I further urge the authors to consider using the absolute flow responses for comparison so that the changes in baseline flow pre to post treatment do not confound the data included in their meta-analyses. I believe this will make the resulting %reduction in the hypoxic blood flow response much more meaningful of an outcome variable.

Major Comment #2: I wonder if the exclusion criteria was too strict, specifically for the human studies. The number of human studies included is quite limited and prohibits meta-analyses. The addition of more human studies would be of benefit to support meta-analyses, otherwise, I think the benefit to readers is limited by the exclusion of many relevant human studies. It may also be worth considering reducing the number of studies you require to perform meta-analyses (currently noted as 4) to allow you to present more information from the human studies. To this point, some clarification regarding why the following studies were not included would be helpful and appreciated (this is not an exhaustive list). It is unclear to me which of the reasons for exclusion at the full text stage these would have been excluded for.

Human studies

Ide, K., Worthley, M., Anderson, T., & Poulin, M. J. (2007). Effects of the nitric oxide synthase inhibitor L-NMMA on cerebrovascular and cardiovascular responses to hypoxia and hypercapnia in humans. *The Journal of Physiology*, 584(1), 321-332.

Hoiland, R. L., Ainslie, P. N., Wildfong, K. W., Smith, K. J., Bain, A. R., Willie, C. K., Foster, G. E., Monteleone, B., & Day, T. A. (2015). Indomethacin-induced impairment of regional cerebrovascular reactivity: implications for respiratory control. *J Physiol*, 593(5), 1291-1306. <https://doi.org/10.1113/jphysiol.2014.284521>

Peltonen, G. L., Harrell, J. W., Rousseau, C. L., Ernst, B. S., Marino, M. L., Crain, M. K., & Schrage, W. G. (2015). Cerebrovascular regulation in men and women: stimulus-specific role of cyclooxygenase. *Physiological Reports*, 3(7), e12451. <https://doi.org/10.14814/phy2.12451>

Harrell, J. W., Peltonen, G. L., & Schrage, W. G. (2019). Reactive oxygen species and cyclooxygenase products explain the majority of hypoxic cerebral vasodilation in healthy humans. *Acta Physiologica*, 226(4), e13288. <https://doi.org/10.1111/apha.13288>

** the 2019 Harrell study could presumably be used for the control hypoxia versus combined ascorbic acid+indomethacin hypoxic response

Animal studies

Takuwa, H., Matsuura, T., Bakalova, R., Obata, T., & Kanno, I. (2010). Contribution of nitric oxide to cerebral blood flow regulation under hypoxia in rats. *Journal of Physiological Sciences*, 60(6), 399-406. <https://doi.org/10.1007/s12576-010-0108-9>

Minor Comment #1: Could the authors please indicate if the systematic review was pre-registered, including the database and registration number. Otherwise, it must be stated that the review was not registered.

Minor Comment #2: Could the authors please speak to why hypoxia must be the first experimental condition following the application of a vehicle/treatment to be included?

Minor Comment #3: Why did the authors use the transcranial Doppler ultrasound data from the Hoiland 2017 and 2023 publications rather than the Duplex ultrasound measures (I see TCD is listed in Supplemental table 4).

Minor Comment #4: I believe the methods are missing a description of statistical analyses.

END OF COMMENTS

Confidential Review

25-Aug-2023

Manuscript ID: JP-TR-2023-285060
Responses to the referees' comments

Reviewing Editor:

This systemic review examined clinical and experimental studies that explored the mechanisms associated with cerebral blood flow regulation during hypoxia. Overall, the article is well-written and brings helpful information to the field. The review underwent evaluation by two experts who expressed enthusiasm and optimism about its content. Additionally, the referees provided relevant feedback for the authors to consider in relation to the analyzed data. These insights have the potential to enhance the review's impact and provide additional valuable information.

We would like to thank both Reviewers and the Editors of *The Journal of Physiology* for their time taken to evaluate our submission, and overall positive assessment of our work. We are grateful for the detailed and constructive comments provided and delighted to have an opportunity to re-submit our work. We now submit a revised manuscript and provide a full response to all the criticisms raised, citing changes to the text of the manuscript where relevant.

Referee #1:

This is a nice systematic review of the literature examining papers that have described the cell pathways involved in cerebral blood flow increases to acute hypoxia in mammals, including a small number of human studies. From an initial search of >5000 papers, the authors narrow it down to 34: 28 animal studies and 6 human studies. They find that the adenosine system, including activation of KATP channels shows the most consistent and largest block of hypoxia induced vasodilation, whereas other pathways are either not involved or more minorly contribute. The paper adheres to systematic review standards. The authors carefully track all the main variables that are different between the papers and clearly divulge them so that readers can make informed interpretations, including the type of animal used, the type of anesthesia, ventilation or not, how CBF was measured and other variables too. Importantly, they also track the confidence interval and report the average p values across papers. All papers included needed a p-values less than 0.05 on the primary measure (a percent reduction in hypoxic-induced vasodilation by the experimental manipulation). One minor short coming is that there are not very many papers (only 34), but this is still informative and useful. I have only a few minor points for the authors.

Response: We thank this Reviewer for their time taken to evaluate our submission and very positive assessment of our work. We are very grateful for the constructive comments provided and appreciate the Reviewer's assessment of our analysis as informative and useful.

Critique: It may be worth while for the authors to consider the error in each of the averages taken from these papers. For example, one paper may show a 40% effect with a $p=0.05$, whereas another paper may show an effect of 10% with a $p=0.001$. Not considering the error from each paper produces a bias towards the average value itself and could affect the interpretation. Similarly, one can make an argument based on the sample size, where large sample sizes could be considered more influential. This could be done by taking a weighted average from each study, and/or calculating the average standard deviation across all the data sets from the original papers, and report that error instead of the SD from the averaging the average values from the papers. I am not

saying the authors must do this, but they should run some tests to see if the outcome and interpretation changes.

Response: We agree and thank the Reviewer for making this suggestion. We attempted to re-analyse the data as suggested by taking a weighted average and the average standard deviation across all the data sets from each of the studies, however, it was not possible for every study because some sources included in this analysis did not report the exact p-values based on the percentage change in the magnitude of the hypoxic response. When these studies were excluded and the data extracted from the remaining sources were analysed as suggested by the reviewer, the outcome and interpretation did not change; our analysis suggests that the key mechanism (responsible for ~40% of the response) of hypoxic cerebral vasodilation involves the actions of adenosine and modulation of vascular K_{ATP} channels.

Critique: The authors say: "one study used an inhibitor of the neuronal and endothelial isoforms of NOS - L-NNA in piglets, one study used a neuronal NOS inhibitor 7-NINA in piglets, and one study used guanyl cyclase inhibitor L-NIO in piglets". As far as I understand, there are no highly selective antagonists for nNOS vs eNOS. 7-NINA is even stated to be a general NOS inhibitor by commercial vendors. Also, L-NIO is not a cGMP inhibitor, but a non-selective antagonist of NOS. Perhaps this sentence could be rephrased, stating that several different NOS inhibitors have been tested.

Response: We thank the Reviewer for making this comment and apologize for our inaccurate description of the pharmacological properties of NOS inhibitors used in the selected studies. The text of the manuscript has been revised to read:

"Five studies involved administration of a non-selective nitric oxide synthase (NOS) inhibitor L-NAME in dogs (1 study), piglets (1 study) and rats (3 studies), one study used an inhibitor of the neuronal and endothelial isoforms of NOS - L-NNA in piglets, one study used NOS inhibitor 7-NINA in piglets, and one study used NOS inhibitor L-NIO also in piglets."

Critique: In the figures, everything is black and white and all the symbols are the same (circles) and it is difficult to follow which circles are connect to which variables/treatments. At least make each group a different symbol: square, triangle, filled unfilled etc., and each text label could also have that corresponding symbol beside the text.

Response: We agree and thank the Reviewer for making this suggestion. We revised Figures 2 and 7 accordingly. The remaining Figures are clear in our opinion as they summarize the data obtained in studies which targeted individual signalling pathways.

Critique: Also, especially for fig3 and fig7, because the graph on the right is very tall, vertical dotted lines are needed for the x-axis ticks, such as 50, 100 etc. It is difficult to see where the points lay without this.

Response: We agree and revised the figures accordingly.

Referee #2:

I very much enjoyed reading this systematic review and meta-analysis. The mechanisms of hypoxic cerebral vasodilation is one of the areas of research I am most passionate

about and I believe the author's have produced an important piece of literature for this field. I hope that the authors find my comments helpful. Sincerely, Ryan Hoiland.

Response: We thank Dr Hoiland for reviewing our submission and overall positive assessment of our work. Please review our detailed responses to all the critical comments raised.

Major Comment #1: One major comment is in regard to the data that is extracted and used to represent the change in CBF during hypoxia. For this I have the following comments/concerns:

1) Can the authors specify if they were comparing the %reduction in the absolute change in CBF (i.e., delta CBF), or the slope response of the change in CBF (i.e., delta CBF / delta SaO₂). It is unclear to me exactly what the extracted data was, although I am fairly certain the authors used the absolute change in CBF in their analyses (e.g., data in Figure 2). There are important implications for considering the variation in stimulus magnitude that can occur between experimental trials within a study, as well as between studies - without taking this into consideration, it is difficult to interpret the data. For example, it is unclear if the difference in CBF in the different subgroups that are reported in figure 2 are due to variation in stimulus magnitude. By reporting these data as the slope of the response (i.e., cerebrovascular reactivity to hypoxia) it would make direct comparisons easier and the data more informative. I urge the authors to consider this as a more appropriate outcome variable.

Response: The data reporting the expression of the cerebrovascular response to hypoxia (in % change from baseline) were extracted from primary sources and used for the analysis. We completely agree with Dr Hoiland that variations in stimulus magnitude should be taken into the account and, ideally, the data should be reported as the slope of the cerebrovascular response to hypoxia. However, the data reporting the slope of the response or data that would be required to calculate the slope are not available for the majority of the experimental animal studies. Many publications simply state that hypoxia was induced by reduction in the level of inspired oxygen (in most cases to 10%), but the effects of the experimental stimulus on SaO₂ and/or PaO₂ were not reported. Therefore, between-studies comparisons of the responses and the effects of treatments normalised to the stimulus magnitude are not possible for the purpose of this analysis. In the revised manuscript we now discuss this important issue as one of the limitations of our analysis.

2) The fact that %changes in CBF are dependent upon the baseline blood flow, which could be influenced by any of the pharmacologic / genetic interventions included, also raises issues for interpreting the data. For example, the change in absolute flow response for the Bowton 1988 study is approximately +50% (8.5mL/100g/min vs. 12.4mL/100g/min), whereas the change in the %change in flow is inflated because of the lower baseline blood flow (the +113.9% you report in Figure 3). This is similar to the Pelligrino 1993 study used in the nitric oxide meta-analysis. Conversely, the 24% reduction reported in the Hoiland 2023 study, is based on the change in the absolute flow response, not the %change in flow. To calculate the %change in the CBF response following a drug based on both the %flow response and absolute flow response, with no indication for when one or the other is being used is confusing and hinders the level to which the data can be compared and subsequently interpreted. I urge the authors add the control CBF response (absolute CBF change scaled to a stimulus variable such as SaO₂ or CaO₂), the intervention CBF response, and the units of measure (e.g., mL/100g/min, cm/s, etc.) to the tables they include with their forest plots. I further urge the authors to consider using the absolute flow responses for comparison so that the changes in baseline flow pre to post treatment do not confound the data included in their

meta-analyses. I believe this will make the resulting %reduction in the hypoxic blood flow response much more meaningful of an outcome variable.

Response: We completely agree and thank the Reviewer for making this suggestion. From 34 studies included in the analysis, 19 reported data as changes in absolute measures of cerebrovascular reactivity (flow, vessel diameter, etc); and 15 studies only reported the normalized data. 19 studies described the effect of the experimental treatment on the baseline variable; whilst the remaining 15 studies did not mention or report changes in baseline cerebrovascular flow following pharmacological treatment or genetic manipulation. To address this comment of the reviewer we re-analysed and re-plotted the data as follows: 1) all human studies included in the analysis provide absolute values; for the analysis/comparison of data obtained in human studies, the data were extracted and the effect of an experimental treatment on the expression of the cerebrovascular response to hypoxia was calculated based on the absolute values and expressed as the percentage of the control response (following the Reviewer's example above); 2) in the majority of animal studies included in the analysis the absolute values were not reported and there were no mentioning in the text of the baseline changes; for the analysis and comparison of data obtained in animal studies, the effects of experimental treatments on the expression of the hypoxic response were calculated based on the reported normalized data; 3) if the effect of the experimental treatment on baseline flow was described or data reported could have been used to assess the differences in the baseline flow, the data were included in the forest plots (as suggested by the Reviewer). These further details of data analysis are now included in the Methods section of the revised manuscript.

Major Comment #2: I wonder if the exclusion criteria was too strict, specifically for the human studies. The number of human studies included is quite limited and prohibits meta-analyses. The addition of more human studies would be of benefit to support meta-analyses, otherwise, I think the benefit to readers is limited by the exclusion of many relevant human studies. It may also be worth considering reducing the number of studies you require to perform meta-analyses (currently noted as 4) to allow you to present more information from the human studies. To this point, some clarification regarding why the following studies were not included would be helpful and appreciated (this is not an exhaustive list). It is unclear to me which of the reasons for exclusion at the full text stage these would have been excluded for.

Response: We thank Dr Hoiland for this comment. Our analysis was performed in accordance with the guidelines (<https://www.ncbi.nlm.nih.gov/books/NBK49407/>) which recommend a minimum of 4 studies for subgroup meta-analysis. In our opinion the exclusion criteria we applied were reasonable to make sure the analysis is not confounded by the data derived from studies in which the experimental design was not sufficiently robust. Because of the small number of human studies targeting the individual pathways, reducing the number of studies required to perform meta-analyses to 3 would not change the outcome and conclusions of this systematic review.

From 246 articles that met the eligibility criteria for full-text screening, 211 articles did not meet the selection criteria for inclusion because of one or more of the following reasons: study protocols did not involve pharmacological or genetic blockade of physiological mechanisms followed by hypoxia testing in vivo (n=126), study protocols did not include control treatments involving the administration of a vehicle (n=35); study protocols did not evaluate the effect of hypoxia on cerebrovascular flow in vivo (n=32), publication type was not an original research article (n=29), non-mammalian species were used (n=3), full text was unavailable (n=3), sample sizes were not reported (n=2), or experimental conditions were as such that hypoxia induced cerebrovascular

constrictions (n=1). Please review below our clarification on why the studies identified by the Reviewer were excluded from the analysis:

Ide, K., Worthley, M., Anderson, T., & Poulin, M. J. (2007). Effects of the nitric oxide synthase inhibitor L-NMMA on cerebrovascular and cardiovascular responses to hypoxia and hypercapnia in humans. *The Journal of Physiology*, 584(1), 321-332. – the study protocol did not include control treatment(s) involving the administration of a vehicle solution.

Peltonen, G. L., Harrell, J. W., Rousseau, C. L., Ernst, B. S., Marino, M. L., Crain, M. K., & Schrage, W. G. (2015). Cerebrovascular regulation in men and women: stimulus-specific role of cyclooxygenase. *Physiological Reports*, 3(7), e12451. – In this study, after the administration of the drug (indomethacin) or placebo treatment, two stimuli - hypoxia and hypercapnia, - were applied sequentially and in random order, therefore, the magnitude of the hypoxic cerebrovascular response might have been affected in the experiments where the CO₂ stimulus was applied 10 min earlier. Yet, the outcome of the study (minimal contribution of COX to the hypoxic cerebrovascular response) is consistent with the results of other studies included in the analysis.

Harrell, J. W., Peltonen, G. L., & Schrage, W. G. (2019). Reactive oxygen species and cyclooxygenase products explain the majority of hypoxic cerebral vasodilation in healthy humans. *Acta Physiologica*, 226(4), e13288. – In this study, hypoxic cerebrovascular responses were studied after administration of a combination of two drugs: ascorbic acid and indomethacin, therefore, the exact target(s) of the experimental treatment applied is unclear.

Takuwa, H., Matsuura, T., Bakalova, R., Obata, T., & Kanno, I. (2010). Contribution of nitric oxide to cerebral blood flow regulation under hypoxia in rats. *Journal of Physiological Sciences*, 60(6), 399-406. – the study protocol did not include control treatment(s) involving the administration of a vehicle.

Hoiland, R. L., Ainslie, P. N., Wildfong, K. W., Smith, K. J., Bain, A. R., Willie, C. K., Foster, G. E., Monteleone, B., & Day, T. A. (2015). Indomethacin-induced impairment of regional cerebrovascular reactivity: implications for respiratory control. *J Physiol*, 593(5), 1291-1306. – Similar to the Peltonen et al (2015) protocol, in this study, after the administration of the drug (indomethacin), two stimuli - hypoxia and hypercapnia - were applied sequentially and in random order; the magnitude of the hypoxic cerebrovascular response might have been affected in experiments where the CO₂ stimulus was applied a few minutes earlier.

Minor Comment #1: Could the authors please indicate if the systematic review was pre-registered, including the database and registration number. Otherwise, it must be stated that the review was not registered.

Response: This systematic review was not pre-registered. We now state this in the revised manuscript.

Minor Comment #2: Could the authors please speak to why hypoxia must be the first experimental condition following the application of a vehicle/treatment to be included?

Response: We reasoned that the expression of the hypoxic cerebrovascular response might be affected by any other experimental manipulation applied between the administration of the test drug and the hypoxic stimulus. For example, from our own experience in animal models we know that altered activity of signalling/biochemical

pathways and physiological responses following systemic hypercapnia would be expected to significantly outlast the duration of the CO₂ stimulus.

Minor Comment #3: Why did the authors use the transcranial Doppler ultrasound data from the Hoiland 2017 and 2023 publications rather than the Duplex ultrasound measures (I see TCD is listed in Supplemental table 4).

Response: We apologize for our error in this description. For our analysis we used the data obtained using the Duplex ultrasound method and reported in the selected publications. The manuscript was revised accordingly.

Minor Comment #4: I believe the methods are missing a description of statistical analyses.

Response: We apologize that in our original submission the description of the statistical analysis was missing. Thank you, we now added this information.

Dear Professor Gourine,

Re: JP-TR-2024-285060R1 "Signalling mechanisms regulating cerebral blood flow during hypoxia" by Alexander Mascarenhas, Alice Braga, Shereen Nizari, Debora Marletta, Shefteeq M Theparambil, Qadeer Aziz, Nephtali Marina, and Alexander V Gourine

Thank you for submitting your manuscript to The Journal of Physiology. It has been assessed by a Reviewing Editor and by 2 expert referees and we are pleased to tell you that it is acceptable for publication following satisfactory revision.

ABSTRACT FIGURES: Authors may use The Journal's premium BioRender account to create/redraw their Abstract Figures (and any other suitable schematic figure). Information on how to access this account is here: <https://physoc.onlinelibrary.wiley.com/journal/14697793/biorender-access>.

REVISION CHECKLIST: Upload a full Response to Referees file. To create your 'Response to Referees' copy all the reports, including any comments from the Senior and Reviewing Editors, into a Microsoft Word, or similar, file and respond to each point, using font or background colour to distinguish comments and responses and upload as the required file type.

We look forward to receiving your revised submission.

Yours sincerely,

EDITOR COMMENTS

Reviewing Editor:

I thank the authors for addressing all the referees' comments and making corresponding changes in the text, which improved the quality of this review article. Referee #2 presented new minor but important comments that require additional consideration from the authors. Moreover, please provide a legend for the Abstract Figure according to the Journal's guidelines (see 'Required Items' below).

REFEREE COMMENTS

Referee #1:

The authors have addressed my comments satisfactorily.

Referee #2:

I would like to thank the authors for the revisions made to their manuscript. As stated in my initial review, I believe this will be a very valuable resource in the field of cerebrovascular physiology, and commend the authors for this work. I have only a few very minor comments, and otherwise would like to congratulate the authors on this work.

Minor comment #1: On page 11 (Methods of cerebrovascular flow measurement), it is noted that 3 studies used transcranial Doppler, and 2 used Duplex ultrasound, and I thank the authors for making the update with regards to duplex ultrasound. However, isn't there 3 studies with Duplex (Hoiland 2017, Rocha 2020, and Hoiland 2023)? Also, in Figure 2C, I can only see 1 data point for Duplex ultrasound. Have two studies been missed in this plot, or do the data for each duplex ultrasound study overlap? I believe the three Duplex ultrasound studies showed different magnitude increases in CBF. Hoiland 2017 had an increase in CBF from 758 to 1245mL/min in the control condition (+64%; "Placebo Pre"), Hoiland 2023 had an increase in CBF from ~800 to ~1075mL/min in the control condition (+35%; "Saline"), while Rocha had an ~35% ("Before PL") increase in the control condition based on figure 3 of their paper. Based on these rough estimations, I believe the Hoiland 2023 paper is missing from Figure 2C, and one of the other studies also?

Also regarding figure 2, I am not 100% sure what data is used in Figure 2A to represent the human data, but there doesn't seem to be a data point that aligns with the aforementioned 64% increase in CBF seen in the Hoiland 2017 paper? There are only 5 visible data points despite the inclusion of 6 human studies.

Minor Comment #2: Apologies if I have missed this, but, if correct, a note indicating that in studies that induced graded stages of hypoxemia the final stage was used may be helpful as an addition description of your methodology. I acknowledge you make the following point, but I am not sure this means the same thing that I have outlined? Page 7 (Data Extraction) "In cases where the experimental data were reported at several timepoints, the largest difference in the hypoxia-induced cerebrovascular response between the control and experimental groups was taken as the effect of the treatment."

Minor Comment #3: For the studies using indomethacin, it is worth considering that indomethacin has been demonstrated to inhibit cAMP dependent protein kinase downstream of prostaglandins effect on cAMP activity, so it is difficult to directly

attribute its influence on cerebral blood flow regulation to reductions in prostaglandin synthesis. Two citations are provided below regarding the influence of Indomethacin on cAMP dependent protein kinase (Kantor & Hampton, 1978; Goueli & Ahmed, 1980).

Goueli SA, Ahmed K. Indomethacin and inhibition of protein kinase reactions. *Nature*. 1980 Sep 11;287(5778):171-2. doi: 10.1038/287171a0. PMID: 6253792.

Kantor HS, Hampton M. Indomethacin in submicromolar concentrations inhibits cyclic AMP-dependent protein kinase. *Nature*. 1978 Dec 21-28;276(5690):841-2. doi: 10.1038/276841a0. PMID: 214715.

While acknowledging the subsequent point falls outside the scope of a review on hypoxic cerebral vasodilation, it is worth noting that Indomethacin is (to my knowledge) the only cyclooxygenase inhibitor that consistently reduces cerebrovascular reactivity to hypercapnia in humans (Markus 1994, *Stroke*, PMID: 8073456; Hoiland 2016, *J Physiol*, PMID: 26880615). If the authors are interested, I had included a table in a Letter I wrote on this topic outlining the different COX inhibitors and their influence on cerebrovascular reactivity to hypercapnia (Hoiland & Ainslie, 2017, *J Physiol*, PMID: 28568772). Unfortunately, a similar amount of data is not available for cerebral blood flow reactivity to hypoxia. With all of this as a bit of a tangential point, I am not sure the author's paper is the right forum to discuss whether or not effects of Indomethacin on the cerebrovascular blood flow response to hypoxia may or may not be prostaglandin dependent, nor do I want such discussion to distract from their excellent work. My one suggestion would simply be to mention somewhere in the discussion that Indomethacin may have non-prostaglandin dependent effects on cerebral blood flow regulation.

Thank you again to the authors for this important contribution.

REQUIRED ITEMS

Please provide a legend to accompany your abstract figure. The abstract figure legend should be included in the article (Word) file, along with the other figure legends.

END OF COMMENTS

1st Confidential Review

22-Mar-2024

Manuscript ID: JP-TR-2023-285060
Responses to the referees' comments

We would like to thank the Reviewers and the Editors of *The Journal of Physiology* for their time taken to evaluate our revised submission and very positive assessment of our work. We are grateful for the additional comments provided. We now submit the second revision of our manuscript and provide our responses to all the remaining comments made by the Reviewers.

Referee #2:

I would like to thank the authors for the revisions made to their manuscript. As stated in my initial review, I believe this will be a very valuable resource in the field of cerebrovascular physiology, and commend the authors for this work. I have only a few very minor comments, and otherwise would like to congratulate the authors on this work.

Response: We thank Dr Hoiland for reviewing our revised submission and very positive assessment of our work. Please review our responses to all the remaining comments made.

Minor comment #1: On page 11 (Methods of cerebrovascular flow measurement), it is noted that 3 studies used transcranial Doppler, and 2 used Duplex ultrasound, and I thank the authors for making the update with regards to duplex ultrasound. However, isn't there 3 studies with Duplex (Hoiland 2017, Rocha 2020, and Hoiland 2023)? Also, in Figure 2C, I can only see 1 data point for Duplex ultrasound. Have two studies been missed in this plot, or do the data for each duplex ultrasound study overlap? I believe the three Duplex ultrasound studies showed different magnitude increases in CBF. Hoiland 2017 had an increase in CBF from 758 to 1245mL/min in the control condition (+64%; "Placebo Pre"), Hoiland 2023 had an increase in CBF from ~800 to ~1075mL/min in the control condition (+35%; "Saline"), while Rocha had an ~35% ("Before PL") increase in the control condition based on figure 3 of their paper. Based on these rough estimations, I believe the Hoiland 2023 paper is missing from Figure 2C, and one of the other studies also?

Also regarding figure 2, I am not 100% sure what data is used in Figure 2A to represent the human data, but there doesn't seem to be a data point that aligns with the aforementioned 64% increase in CBF seen in the Hoiland 2017 paper? There are only 5 visible data points despite the inclusion of 6 human studies.

Response: Thank you very much for this thorough evaluation of our submission. We apologize for this error. The data point obtained from Rocha 2020 publication was included under the "transcranial Doppler ultrasound" category, - now corrected. Also, in our analysis of data obtained in human studies we extracted the values of cerebrovascular flow that were taken at 80% SaO₂/SpO₂ and reported in the selected publications. This allowed comparisons of the response magnitude between different human studies. We now revised the text of the manuscript and explicitly indicate that for the analysis of data from human studies the values obtained at 80% SaO₂ were extracted. Control cerebrovascular responses at 80% saturation reported in studies by Hoiland 2017, Hoiland 2023, and Rocha 2020 were found to be very similar: 33.6%, 34.7%, and 31.59%, respectively. The 64% increase in CBF reported in Hoiland 2017 paper was obtained at 70% SaO₂ (more severe level of hypoxia was applied in this particular study), therefore, we used the CBF values obtained at 80% saturation. We have now added the missing datapoints to revised Figure 2, corrected all the relevant figures, and updated the text of the manuscript accordingly.

Minor Comment #2: Apologies if I have missed this, but, if correct, a note indicating that in studies that induced graded stages of hypoxemia the final stage was used may be helpful as an additional description of your methodology. I acknowledge you make the following point, but I am not sure this means the same thing that I have outlined? Page 7 (Data Extraction) "In cases where the experimental data were reported at several timepoints, the largest difference in the hypoxia-induced cerebrovascular response between the control and experimental groups was taken as the effect of the treatment."

Response: Thank you for this comment. We now revised the text of the manuscript to read:

"In cases where the data obtained in experimental animal studies were reported at several timepoints and at graded stages of hypoxia, the largest difference in the hypoxia-induced cerebrovascular response between the control and experimental groups was taken as the effect of the treatment".

Minor Comment #3: For the studies using indomethacin, it is worth considering that indomethacin has been demonstrated to inhibit cAMP dependent protein kinase downstream of prostaglandins effect on cAMP activity, so it is difficult to directly attribute its influence on cerebral blood flow regulation to reductions in prostaglandin synthesis. Two citations are provided below regarding the influence of Indomethacin on cAMP dependent protein kinase (Kantor & Hampton, 1978; Goueli & Ahmed, 1980).

Goueli SA, Ahmed K. Indomethacin and inhibition of protein kinase reactions. *Nature*. 1980 Sep 11;287(5778):171-2. doi: 10.1038/287171a0. PMID: 6253792.

Kantor HS, Hampton M. Indomethacin in submicromolar concentrations inhibits cyclic AMP-dependent protein kinase. *Nature*. 1978 Dec 21-28;276(5690):841-2. doi: 10.1038/276841a0. PMID: 214715.

While acknowledging the subsequent point falls outside the scope of a review on hypoxic cerebral vasodilation, it is worth noting that Indomethacin is (to my knowledge) the only cyclooxygenase inhibitor that consistently reduces cerebrovascular reactivity to hypercapnia in humans (Markus 1994, *Stroke*, PMID: 8073456; Hoiland 2016, *J Physiol*, PMID: 26880615). If the authors are interested, I had included a table in a Letter I wrote on this topic outlining the different COX inhibitors and their influence on cerebrovascular reactivity to hypercapnia (Hoiland & Ainslie, 2017, *J Physiol*, PMID: 28568772). Unfortunately, a similar amount of data is not available for cerebral blood flow reactivity to hypoxia. With all of this as a bit of a tangential point, I am not sure the author's paper is the right forum to discuss whether or not effects of Indomethacin on the cerebrovascular blood flow response to hypoxia may or may not be prostaglandin dependent, nor do I want such discussion to distract from their excellent work. My one suggestion would simply be to mention somewhere in the discussion that Indomethacin may have non-prostaglandin dependent effects on cerebral blood flow regulation.

Response: We thank Dr Hoiland for making this comment. We are fully aware of the off-target effects of indomethacin and made exactly this point in our 2019 publication (PMID:30481531): *"In humans, systemic non-specific COX inhibition was also reported to reduce the magnitude of the BOLD response induced by visual stimulation by 47% (Bruhn et al., 2001). It is important to note that indomethacin, used in all these studies as a non-specific COX inhibitor, is also a potent inhibitor of cyclic AMP-dependant protein kinase activity (Goueli and Ahmed, 1980), which is integral to any vasomotor response. Therefore, conclusions drawn from the results of the experiments using indomethacin as a COX inhibitor may somewhat overestimate the importance of this pathway. For example, in humans indomethacin was found to reduce cerebrovascular CO₂ reactivity, while the other non-specific COX inhibitors (naproxen, ketorolac) had no effect on CO₂-induced dilations (Hoiland et al., 2016)".*

We now included a brief discussion of this issue in the revised manuscript. Thank you!

Dear Professor Gourine,

Re: JP-TR-2024-285060R2 "On the mechanisms of brain blood flow regulation during hypoxia" by Alexander Mascarenhas, Alice Braga, Sara Maria Majernikova, Shereen Nizari, Debora Marletta, Shefeeq M Theparambil, Qadeer Aziz, Nephtali Marina, and Alexander V Gourine

We are pleased to tell you that your paper has been accepted for publication in The Journal of Physiology.

Authors should note that it is too late at this point to offer corrections prior to proofing. Major corrections at proof stage, such as changes to figures, will be referred to the Editors for approval before they can be incorporated. Only minor changes, such as to style and consistency, should be made at proof stage. Changes that need to be made after proof stage will usually require a formal correction notice.

Yours sincerely,

Laura Bennet
Senior Editor
The Journal of Physiology

P.S. - You can help your research get the attention it deserves! Check out Wiley's free Promotion Guide for best-practice recommendations for promoting your work at www.wileyauthors.com/eeo/guide. You can learn more about Wiley Editing Services which offers professional video, design, and writing services to create shareable video abstracts, infographics, conference posters, lay summaries, and research news stories for your research at www.wileyauthors.com/eeo/promotion.

IMPORTANT NOTICE ABOUT OPEN ACCESS: To assist authors whose funding agencies mandate public access to published research findings sooner than 12 months after publication, The Journal of Physiology allows authors to pay an Open Access (OA) fee to have their papers made freely available immediately on publication.

You can check if your funder or institution has a Wiley Open Access Account here: <https://authorservices.wiley.com/author-resources/Journal-Authors/licensing-and-open-access/open-access/author-compliance-tool.html>.

EDITOR COMMENTS

Reviewing Editor:

I thank the authors for addressing all comments. Congratulations on this insightful manuscript.